# Lookback Prophet Inequalities

**Ziyad Benomar**
ENSAE, Ecole Polytechnique,
FairPlay joint team
ziyad.benomar@ensae.fr

**Dorian Baudry**
Department of Statistics,
University of Oxford
dorian.baudry@ox.ac.uk

**Vianney Perchet**
CREST, ENSAE, Criteo AI LAB
Fairplay joint team
vianney.perchet@normalesup.org

## Abstract

Prophet inequalities are fundamental optimal stopping problems, where a decision-maker observes sequentially items with values sampled independently from known distributions, and must decide at each new observation to either stop and gain the current value or reject it irrevocably and move to the next step. This model is often too pessimistic and does not adequately represent real-world online selection processes. Potentially, rejected items can be revisited and a fraction of their value can be recovered. To analyze this problem, we consider general decay functions $D_1, D_2, \ldots$, quantifying the value to be recovered from a rejected item, depending on how far it has been observed in the past. We analyze how lookback improves, or not, the competitive ratio in prophet inequalities in different order models. We show that, under mild monotonicity assumptions on the decay functions, the problem can be reduced to the case where all the decay functions are equal to the same function $x \mapsto \gamma x$, where $\gamma = \inf_{x>0} \inf_{j \geq 1} D_j(x)/x$. Consequently, we focus on this setting and refine the analyses of the competitive ratios, with upper and lower bounds expressed as increasing functions of $\gamma$.

## 1 Introduction

Optimal stopping problems constitute a classical paradigm of decision-making under uncertainty [Dynkin, 1963] Typically, in online algorithms, these problems are formalized as variations of the secretary problem [Lindley, 1961] or the prophet inequality [Krengel and Sucheston, 1977]. In the context of the prophet inequality, the decision-maker observes a finite sequence of items, each having a value drawn independently from a known probability distribution. Upon encountering a new item, the decision-maker faces the choice of either accepting it and concluding the selection process or irreversibly rejecting it, with the objective of maximizing the value of the selected item. However, while the prophet inequality problem is already used in scenarios such as posted-price mechanism design [Hajiaghayi et al., 2007] or online auctions [Syrgkanis, 2017], it might present a pessimistic model of real-world online selection problems. Indeed, it is in general possible in practice to revisit previously rejected items and potentially recover them or at least recover a fraction of their value.

Consider for instance an individual navigating a city in search of a restaurant. When encountering one, they have the choice to stop and dine at this place, continue their search, or revisit a previously passed option, incurring a utility cost that is proportional to the distance of backtracking. In another example drawn from the real estate market, homeowners receive offers from potential buyers. The decision to accept or reject an offer can be revisited later, although buyer interest may have changed, resulting in a potentially lower offer or even a lack of interest. Lastly, in the financial domain, an

38th Conference on Neural Information Processing Systems (NeurIPS 2024).

agent may choose to sell an asset at its current price or opt for a lookback put option, allowing them to sell at the asset's highest price over a specified future period. To make a meaningful comparison between the two, one must account for factors such as discounting (time value of money) and the cost of the option.

## 1.1 Formal problem and notation

To encompass diverse scenarios, we propose a general way to quantify the cost incurred by the decision-maker for retrieving a previously rejected value.

**Definition 1.1** (Decay functions). *Let $\mathcal{D} = (D_1, D_2, \ldots)$ be a sequence of non-negative functions defined on $[0, \infty)$. It is a sequence of decay functions if*

    *(i) $D_1(x) \leq x$ for all $x \geq 0$,*

    *(ii) the sequence $(D_j(x))_{j \geq 1}$ is non-increasing for all $x \geq 0$,*

    *(iii) the function $D_j$ is non-decreasing for all $j \geq 1$.*

In the context of decay functions $\mathcal{D}$, if a value $x$ is rejected, the algorithm can recover $D_j(x)$ after $j$ subsequent steps. The three conditions defining decay functions serve as fundamental prerequisites for the problem. The first and second conditions ensure that the recoverable value of a rejected item can only diminish over time, while the final condition implies that an increase in the observed value $x$ corresponds to an increase in the potential recovered value. Although the non-negativity of the decay functions is non-essential, we retain it for convenience, as we can easily revert to this assumption by considering that the algorithm has a reward of zero by not selecting any item.

The motivating examples that we introduced can be modeled respectively with decay functions of the form $D_j(x) = x - c_j$ where $(c_j)_{j \geq 1}$ is a non-decreasing positive sequence, $D_j(x) = \xi_j x$ with $\xi_j \sim \mathcal{B}(p_j)$ and $(p_j)_{j \geq 1}$ a non-increasing sequence of probabilities, and $D_j(x) = \lambda^j x$ with $\lambda \in [0, 1]$. In one of these examples (housing market), the natural model is to use *random decay functions*: the buyer makes the same offer if they are still interested, and offers 0 otherwise. Definition 1.1 can be easily extended to consider this case. However, to enhance the clarity of the presentation, we only discuss the deterministic case in the rest of the paper. In Appendix D, we explain how all the proofs and theorems can be generalized to that case.

**The $\mathcal{D}$-prophet inequality.** For any decay functions $\mathcal{D}$, we define the $\mathcal{D}$-prophet inequality problem, where the decision maker, knowing $\mathcal{D}$, observes sequentially the values $X_1, \ldots, X_n$, with $X_i$ drawn from a known distribution $F_i$ for all $i \in [n]$. If they decide to stop at some step $\tau$, then instead of gaining $X_\tau$ as in the classical prophet inequality, they can choose to select the current item $X_\tau$ and have its full value, or select any item $X_i$ with $i < \tau$ among the rejected ones but only recover a fraction $D_{\tau-i}(X_i) \leq X_i$ of its value. Therefore, if an algorithm ALG stops at step $\tau$ its reward is

$$\mathsf{ALG}^{\mathcal{D}}(X_1, \ldots, X_n) = \max\{X_\tau, D_1(X_{\tau-1}), D_2(X_{\tau-2}), \ldots, D_{\tau-1}(X_1)\}$$
$$= \max_{0 \leq i \leq \tau-1}\{D_i(X_{\tau-i})\},$$

with the convention $D_0(x) = x$. If the algorithm does not stop at any step before $n$, then its reward is $\mathsf{ALG}^{\mathcal{D}}(X_1, \ldots, X_n) = \max_{1 \leq i \leq n}\{D_i(X_{\pi(n-i+1)})\}$, which corresponds to $\tau = n + 1$.

**Remark 1.1.** *As in the standard prophet inequality, an algorithm is defined by its stopping time, i.e., the rule set to decide whether to stop or not. Hence, if $\mathcal{D}$ and $\mathcal{D}'$ are two different sequences of decay functions, any algorithm for the $\mathcal{D}$-prophet inequality, although its stopping time might depend on the particular sequence of functions $\mathcal{D}$, is also an algorithm for the $\mathcal{D}'$-prophet inequality. Consider for example an algorithm ALG with stopping time $\tau(\mathcal{D})$ that depends on $\mathcal{D}$. Its reward in the $\mathcal{D}'$-prophet inequality is $\mathsf{ALG}^{\mathcal{D}'}(X_1, \ldots, X_n) = \max_{0 \leq i \leq \tau-1}\{D'_i(X_{\tau(\mathcal{D})-i})\}$.*

**Observation order.** Several variants of the prophet inequality problem have been studied, depending on the order of observations. The standard model is the adversarial (or fixed) order: The instance of the distributions $F_1, \ldots, F_n$ is chosen by an adversary, and the algorithm observes the samples $X_1 \sim F_1, \ldots, X_n \sim F_n$ in this order [Krengel and Sucheston, 1977, 1978]. In the *random order* model, the adversary can again choose the distributions, but the algorithm observes the samples in a

uniformly random order. Another setting in which the observation order is no longer important is the IID model [Hill and Kertz, 1982, Correa et al., 2021b], where all the values are sampled independently from the same distribution $F$. The $\mathcal{D}$-prophet inequality is well-defined in each of these different order models: if the items are observed in the order $X_{\pi(1)}, \ldots, X_{\pi(n)}$ with $\pi$ a permutation of $[n]$, then the reward of the algorithm is $\mathsf{ALG}^{\mathcal{D}}(X_1, \ldots, X_n) = \max_{0 \leq i \leq \tau-1}\{D_i(X_{\pi(\tau-i)})\}$. In this paper, we study the $\mathcal{D}$-prophet inequality in the three models we presented, providing lower and upper bounds in each of them.

**Competitive ratio.** In the $\mathcal{D}$-prophet inequality, an input instance $I$ is a finite sequence of probability distributions $(F_1, \ldots, F_n)$. Thus, for any instance $I$, we denote by $\mathbb{E}[\mathsf{ALG}^{\mathcal{D}}(I)]$ the expected reward of $\mathsf{ALG}$ given $I$ as input, and we denote by $\mathbb{E}[\mathsf{OPT}(I)]$ the expected maximum of independent random variables $(X_i)_{i \in [n]}$, where $X_i \sim F_i$. With these notations, we define the competitive ratio, which will be used to measure the quality of the algorithms.

**Definition 1.2** (Competitive ratio). *Let $\mathcal{D}$ be a sequence of decay functions and $\mathsf{ALG}$ an algorithm. We define the competitive ratio of $\mathsf{ALG}$ by*

$$CR^{\mathcal{D}}(\mathsf{ALG}) = \inf_I \frac{\mathbb{E}[\mathsf{ALG}^{\mathcal{D}}(I)]}{\mathbb{E}[\mathsf{OPT}(I)]} ,$$

*with the infimum taken over the tuples of all sizes of non-negative distributions with finite expectation.*

An algorithm is said to be $\alpha$-competitive if its competitive ratio is at least $\alpha$, which means that for any possible instance $I$, the algorithm guarantees a reward of at least $\alpha\mathbb{E}[\mathsf{OPT}(I)]$. The notion of competitive ratio is used more broadly in competitive analysis as a metric to evaluate online algorithms [Borodin and El-Yaniv, 2005].

## 1.2 Contributions

It is trivial that non-zero decay functions $\mathcal{D}$ guarantee a better reward compared to the classical prophet inequality. However, in general, this is not sufficient to conclude that the standard upper bounds or the competitive ratio of a given algorithm can be improved. Hence, a first key question is: what condition on $\mathcal{D}$ is necessary to surpass the conventional upper bounds of the classical prophet inequality? Surprisingly, the answer hinges solely on the constant $\gamma_{\mathcal{D}}$, defined as follows,

$$\gamma_{\mathcal{D}} = \inf_{x>0} \inf_{j \geq 1} \left\{ \frac{D_j(x)}{x} \right\} . \tag{1}$$

In the adversarial order model, we demonstrate that the optimal competitive ratio achievable in the $\mathcal{D}$-prophet inequality is determined by the parameter $\gamma_{\mathcal{D}}$ alone. Additionally, in both the random order and IID models, we demonstrate the essential requirement of $\gamma_{\mathcal{D}} > 0$ for breaking the upper bounds of the classical prophet inequality. In particular, this implies that no improvement can be made with decay functions of the form $D_j(x) = x - c_j$ with $c_j > 0$, or $D_j(x) = \lambda^j x$ with $\lambda \in [0, 1)$. Subsequently, we develop algorithms and provide upper bounds in the $\mathcal{D}$-prophet inequality, uniquely dependent on the parameter $\gamma_{\mathcal{D}}$. We illustrate them in Figure 1, comparing them with the identity function $\gamma \mapsto \gamma$, which is a trivial lower bound.

## 1.3 Related work

**Prophet inequalities.** The first prophet inequality was proven by Krengel and Sucheston [Krengel and Sucheston, 1977, 1978] in the setting where the items are observed in a fixed order, demonstrating that the dynamic programming algorithm has a competitive ratio of $1/2$, which is the best possible. It was shown later that the same guarantee can be obtained with simpler algorithms [Samuel-Cahn, 1984, Kleinberg and Weinberg, 2012], accepting the first value above a carefully chosen threshold. For a more comprehensive and historical overview, we refer the interested reader to surveys on the problem such as [Lucier, 2017, Correa et al., 2019]. Prophet inequalities have immediate applications in mechanism design [Hajiaghayi et al., 2007, Deng et al., 2022, Psomas et al., 2022, Makur et al., 2024], auctions [Syrgkanis, 2017, Dütting et al., 2020], resource management [Sinclair et al., 2023], and online matching [Cohen et al., 2019, Ezra et al., 2020, Jiang et al., 2021, Papadimitriou et al., 2021, Brubach et al., 2021]. Many variants and related problems have been studied, including, for example, the matroid prophet inequality [Kleinberg and Weinberg, 2012, Feldman et al., 2016],

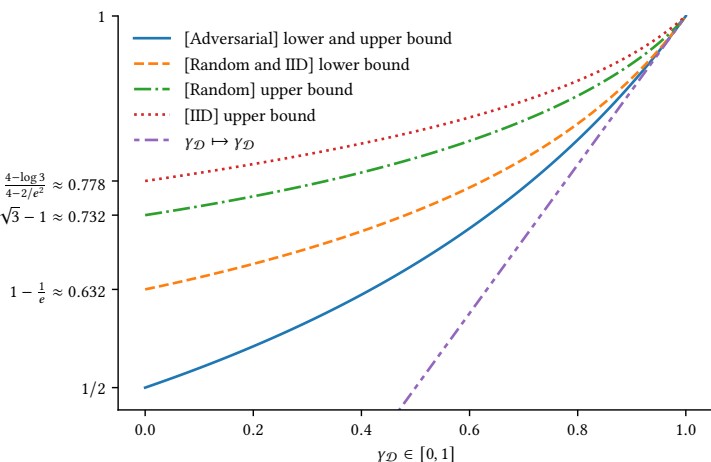

Figure 1: Lower and upper bounds on the competitive ratio in the $\mathcal{D}$-prophet inequality depending on $\gamma_{\mathcal{D}}$, in the adversarial order (Thm 4.3), random order (Thm 4.4) and IID (Thm 4.6) models

prophet inequality with advice [Diakonikolas et al., 2021], and variants with fairness considerations [Correa et al., 2021a, Arsenis and Kleinberg, 2022].

**Random order and IID models.** Esfandiari et al. [2017] introduced the *prophet secretary* problem, where items are observed in a uniformly random order, and they proved a $(1 - \frac{1}{e})$-competitive algorithm. Correa et al. [2021c] showed later a competitive ratio of $0.669$, and Harb [2024] enhanced it to $0.6724$, which currently stands as the best-known solution for the problem. They also proved an upper bound of $\sqrt{3} - 1 \approx 0.732$, which was improved to $0.7254$ in [Bubna and Chiplunkar, 2023] then $0.723$ in [Giambartolomei et al., 2023]. Addressing the gap between the lower and upper bound remains an engaging and actively pursued open question. On the other hand, the study of prophet inequalities with IID random variables dates back to papers such as [Hill and Kertz, 1982, Kertz, 1986], demonstrating guarantees on the dynamic programming algorithm. The problem was completely solved in [Correa et al., 2021b], where the authors show that the competitive ratio of the dynamic programming algorithm is $0.745$, thus it constitutes an upper bound on the competitive ratio of any algorithm, and they give a simpler adaptive threshold algorithm matching it. Another setting that we do not study in this paper, is the *order selection* model, where the decision-maker can choose the order in which the items are observed, knowing their distributions [Chawla et al., 2010, Beyhaghi et al., 2021, Peng and Tang, 2022].

**Beyond the worst-case.** In recent years, there has been increasing interest in exploring ways to exceed the worst-case upper bounds of online algorithms by providing the decision-maker with additional capabilities. A notable research avenue is learning-augmented algorithms [Lykouris and Vassilvtiskii, 2018], which equip the decision-maker with predictions or hints about unknown variables of the problem. Multiple problems have been studied in this framework, such as scheduling [Purohit et al., 2018, Lassota et al., 2023, Benomar and Perchet, 2024b], matching [Antoniadis et al., 2020, Dinitz et al., 2021, Chen et al., 2022], caching [Antoniadis et al., 2023, Chlkedowski et al., 2021, Christianson et al., 2023], the design of data structures [Kraska et al., 2018, Lin et al., 2022, Benomar and Coester, 2024], and in particular, online selection problems [Dütting et al., 2021, Sun et al., 2021, Benomar et al., 2023, Benomar and Perchet, 2024a, Diakonikolas et al., 2021]. More related to our setting, the ability to revisit items in online selection has been studied in problems such as the multiple-choice prophet inequality, where the algorithm can select up to $k$ items and its reward is the maximum selected value [Assaf and Samuel-Cahn, 2000]. This allows for revisiting up to $k$ items, chosen during the execution, for final acceptance or rejection decisions. Similarly, in Pandora's box problem [Weitzman, 1978, Kleinberg et al., 2016] and its variants [Esfandiari et al., 2019, Gergatsouli and Tzamos, 2022, Atsidakou et al., 2024, Gergatsouli and Tzamos, 2024, Berger et al., 2024], the decision maker decides the observation order of the items, but a cost $c_i$ is paid for observing each value $X_i$, with the gain being the maximum observed value minus the total opening costs. A very recent work investigates a scenario closely related to the lookback prophet inequality

[Ekbatani et al., 2024] where, upon selecting a candidate $X_i$, the decision-maker has the option to discard it and choose a new value $X_j$ at any later step $j$, at a buyback cost of $fX_i$, where $f > 0$. The authors present an optimal algorithm for the case when $f \geq 1$, although the problem remains open for $f \in (0, 1)$. Other problems were studied in similar settings, such as online matching [Ekbatani et al., 2022] and online resource allocation [Ekbatani et al., 2023].

## 2 From $\mathcal{D}$-prophet to the $D_\infty$-prophet inequality

Let us consider a sequence $\mathcal{D}$ of decay functions. By Definition 1.1, for any $x \in [0, \infty]$ the sequence $(D_j(x))_{j \geq 1}$ converges, since it is non-increasing and non-negative. Hence, there exists a mapping $D_\infty$ such that for any $x \geq 0$, $\lim_{j \to \infty} D_j(x) = D_\infty(x)$. Furthermore, we can easily verify that $D_\infty$ is non-decreasing and satisfies $D_\infty(x) \in [0, x]$ for all $x \geq 0$.

Thanks to these properties, we obtain that $(D_\infty)_{j \geq 1}$ also satisfies Definition 1.1, and is hence a valid sequence of decay functions. We thus refer to the corresponding problem as the $D_\infty$-prophet inequality. Since $D_j \geq D_\infty$ for any $j \geq 1$, it is straightforward that the stopping problem with the decay functions $D_\infty$ would be less favorable to the decision-maker. More precisely, for any random variables $X_1, \dots, X_n$, observation order $\pi$, and algorithm $\mathsf{ALG}$ with stopping time $\tau$, it holds that

$$\mathsf{ALG}^{\mathcal{D}}(X_1, \dots, X_n) := \max\{X_{\pi(\tau)}, \max_{i < \tau} D_{\tau - i}(X_{\pi(i)})\} \geq \max\{X_{\pi(\tau)}, \max_{i < \tau} D_\infty(X_{\pi(i)})\} \,,$$

which corresponds to the output of $\mathsf{ALG}$ (with the same decision rule) when all the decay functions are equal to $D_\infty$. Therefore, any guarantees established for algorithms in the $D_\infty$-prophet inequality naturally extend to the $\mathcal{D}$-prophet inequality. However, it remains uncertain whether the $\mathcal{D}$-prophet inequality can yield improved competitive ratios compared to the $D_\infty$-prophet inequality. In the following, we prove that this is not the case, for all the order models presented in Section 1.

**Theorem 2.1.** *Let $D_\infty$ be the pointwise limit of the sequence of decay functions $\mathcal{D} = (D_j)_{j \geq 1}$. Then for any instance $I = (F_1, \dots, F_n)$ of non-negative distributions, it holds in the adversarial and the random order models that*

$$\forall \mathsf{ALG} : \mathsf{CR}^{\mathcal{D}}(\mathsf{ALG}) \leq \sup_{\mathsf{A}} \frac{\mathbb{E}[\mathsf{A}^{D_\infty}(I)]}{\mathbb{E}[\mathsf{OPT}(I)]} \,, \tag{2}$$

*where the supremum is taken over all the online algorithms $\mathsf{A}$. In the IID model, the same inequality holds with an additional $O(n^{-1/3})$ term, which depends only on the size $n$ of the instance.*

The main implication of Theorem 2.1 is the following corollary.

**Corollary 2.1.1.** *In the adversarial order and the random order models, if $\bar{A}_\infty$ is an optimal algorithm for the $D_\infty$-prophet inequality, i.e. with maximal competitive ratio, then $\bar{A}_\infty$ is also optimal for the $\mathcal{D}$-prophet inequality. Moreover, it holds that*

$$\mathsf{CR}^{\mathcal{D}}(\bar{A}_\infty) = \mathsf{CR}^{D_\infty}(\bar{A}_\infty) \,.$$

A direct consequence of this result is that, in the adversarial and the random order models, the asymptotic decay $D_\infty$ entirely determines the competitive ratio that is achievable and the upper bounds for the $\mathcal{D}$-prophet inequality. Therefore, we can restrict our analysis to algorithms designed for the problem with identical decay function. In the IID model, the same conclusion holds if the worst-case instances are arbitrarily large, making the additional $O(n^{-1/3})$ term vanish. This is the case in particular in the classical IID prophet inequality [Hill and Kertz, 1982].

### 2.1 Sketch of the proof of Theorem 2.1

While we use different techniques for each order model considered, all the proofs share the same underlying idea. Given any instance $I$ of non-negative distributions, we build an alternative instance $J$ such that the reward of any algorithm on $I$ with decay functions $\mathcal{D} = (D_j)_j$ is at most its reward on $J$ with decay functions all equal to $D_\infty$. To do this, we essentially introduce an arbitrarily large number of zero values between two successive observations drawn from distributions belonging to $I$. Hence, under $J$, the algorithm cannot recover much more than a fraction $D_\infty(X)$ for any past observation $X$ collected from a distribution $F \in I$.

In the adversarial case, implementing this idea is straightforward, since nature can build $J$ by directly inserting $m$ zeros between each pair of consecutive values, and the result is obtained by making $m$ arbitrarily large. For the random order model, we use the same instance $J$, but extra steps are needed to prove that the number of steps between two non-zero values is very large with high probability.

Moving to the IID model, an instance $I$ is defined by a pair $(F, n)$, where $F$ is a non-negative distribution, and $n$ is the size of the instance. In this scenario, we consider an instance consisting of $m > n$ IID random variables $(Y_i)_{i \in [m]}$, each sampled from $F$ with probability $n/m$, and equal to zero with the remaining probability. We again achieve the desired result by letting $m$ be arbitrarily large compared to $n$. However, the number of variables sampled from $F$ is not fixed; it follows a Binomial distribution with parameters $(m, n/m)$. We control this variability by using concentration inequalities, which causes the additional term $O(n^{-1/3})$.

## 3 From $D_\infty$-prophet to the $\gamma_\mathcal{D}$-prophet inequality

As discussed in Section 2, Theorem 2.1 implies that, for either establishing upper bounds or guarantees on the competitive ratios of algorithms, it is sufficient to study the $D_\infty$-prophet inequality, where all the decay functions are equal to $D_\infty$. The remaining question is then to determine which functions $D_\infty$ allow to improve upon the upper bounds of the classical prophet inequality. Before tackling this question, let us make some observations regarding algorithms in the $D_\infty$-prophet inequality.

In the $D_\infty$-prophet inequality, it is always possible to have a reward of $D_\infty(\max_{i \in [n]} X_i)$ by rejecting all the items and then selecting the maximum by the end. Thus, it is suboptimal to stop at a step $i$ where $X_i \leq D_\infty(\max_{j < i} X_j)$. An algorithm respecting this decision rule is called *rational*.

**Lemma 3.1.** *For any rational algorithm* ALG *in the $D_\infty$-prophet inequality, if we denote by $\tau$ its stopping time, then for any instance $I = (F_1, \ldots, F_n)$ and $X_i \sim F_i$ for all $i \in [n]$ we have*

$$\mathsf{ALG}^{D_\infty}(X_1, \ldots, X_n) = \mathsf{ALG}^0(X_1, \ldots, X_n) + D_\infty\big(\max_{i \in [n]} X_i\big) \mathbb{1}_{\tau = n+1} \, ,$$

*where* $\mathsf{ALG}^0$ *denotes the reward of the algorithm in the standard prophet inequality. Moreover, the optimal dynamic programming algorithm in the $D_\infty$-prophet inequality is rational.*

The best competitive ratio in the $D_\infty$-prophet inequality is achieved, possibly among others, by the optimal dynamic programming algorithm, which is a rational algorithm by the previous Lemma. Hence, it suffices to prove upper bounds on rational algorithms. We use this observation to prove the next propositions.

**Proposition 3.2.** *In the $D_\infty$-prophet inequality, if $\inf_{x>0} \frac{D_\infty(x)}{x} = 0$, then it holds, in any order model, that*

$$\forall \mathsf{ALG} : \mathsf{CR}^{D_\infty}(\mathsf{ALG}) \leq \sup_A \mathsf{CR}^0(A) \, , \tag{3}$$

*where the supremum is taken over all the online algorithms $A$, and $\mathsf{CR}^0$ denotes the competitive ratio in the standard prophet inequality.*

Proposition 3.2 implies that if $\inf_{x>0} \frac{D_\infty(x)}{x} = 0$, then, in any order model, any upper bound on the competitive ratios of all algorithms in the classical prophet inequality is also an upper bound on the competitive ratios of all algorithm in the $D_\infty$-prophet inequality. Consequently, for surpassing the upper bounds of the classical prophet inequality, it is necessary to have, for some $\gamma > 0$, that $D_\infty(x) \geq \gamma x$ for all $x \geq 0$. Furthermore, the next Proposition allows giving upper bounds in the $D_\infty$-prophet inequality that depend only on $\inf_{x>0} \frac{D_\infty(x)}{x}$.

**Proposition 3.3.** *Let $\gamma = \inf_{x>0} D_\infty(x)/x$, and $0 < a < b$. Consider an instance $I$ of distributions with support in $\{0, a, b\}$, then in any order model and for any algorithm* ALG *we have that*

$$\mathsf{CR}^{D_\infty}(\mathsf{ALG}) \leq \sup_A \frac{\mathbb{E}[A^\gamma(I)]}{\mathbb{E}[\mathsf{OPT}(I)]} \, ,$$

*where $\mathbb{E}[A^\gamma(I)]$ is the reward of $A$ if all the decay functions were equal to $x \mapsto \gamma x$.*

The core idea for proving this proposition is that rescaling an instance, i.e. considering $(rX_i)_{i \in [n]}$ instead of $(X_i)_{i \in [n]}$, has no impact in the classical prophet inequality. However, in the $D_\infty$-prophet

inequality, rescaling can be exploited to adjust the ratio $\frac{D_\infty(rx)}{rx}$. By considering instances with random variables taking values in $\{0, a, b\}$ almost surely, where $a < b$, a reasonable algorithm facing such an instance would never reject the value $b$. Consequently, the value it recovers from rejected items is either $D_\infty(0) = 0$ or $D_\infty(a)$. Rescaling this instance by a factor $r = s/a$ and taking the ratio to the expected maximum, the term $\frac{D_\infty(s)}{s}$ appears, with $s$ a free parameter that can be chosen to satisfy $\frac{D_\infty(s)}{s} \to \inf_{x>0} \frac{D_\infty(x)}{x} = \gamma_\mathcal{D}$.

As a consequence, if $\inf_{x>0} \frac{D_\infty(x)}{x} = \gamma$, then any upper bound obtained in the $\gamma$-prophet inequality (when the decay functions are all equal to $x \mapsto \gamma x$) using instances of random variables $(X_1, \ldots, X_n)$ satisfying $X_i \in \{0, a, b\}$ a.s. for all $i$, is also an upper bound in the $D_\infty$-prophet inequality.

**Implication**   Consider any sequence $\mathcal{D}$ of decay functions, and define

$$\gamma_\mathcal{D} := \inf_{x>0} \left\{ \frac{D_\infty(x)}{x} \right\} = \inf_{x>0} \inf_{j \geq 1} \left\{ \frac{D_j(x)}{x} \right\} .$$

For any $x > 0$ and $j \geq 1$ it holds that $D_j(x) \geq \gamma_\mathcal{D} x$, therefore, any guarantees on the competitive ratio of an algorithm in the $\gamma_\mathcal{D}$-prophet inequality are valid in the $\mathcal{D}$-prophet inequality, under any order model. Furthermore, combining Theorem 2.1 and Proposition 3.3, we obtain that for any instance $I$ of random variables taking values in a set $\{0, a, b\}$ it holds that

$$\forall \mathsf{ALG} : \mathsf{CR}^\mathcal{D}(\mathsf{ALG}) \leq \sup_\mathsf{A} \frac{\mathbb{E}[\mathsf{A}^{\gamma_\mathcal{D}}(I)]}{\mathbb{E}[\mathsf{OPT}(I)]} ,$$

with an additional term of order $O(n^{-1/3})$ in the IID model. In the particular case where $\gamma_\mathcal{D} = 0$, Proposition 3.3 with Theorem 2.1 give a stronger result, showing that no algorithm can surpass the upper bounds of the classical prophet inequality. This is true also for the IID model since the instances used to prove the tight upper bound of $\approx 0.745$ are of arbitrarily large size [Hill and Kertz, 1982].

Therefore, by studying the $\gamma$-prophet inequality for $\gamma \in [0, 1]$, we can prove upper bounds and lower bounds on the $\mathcal{D}$-prophet inequality for any sequence $\mathcal{D}$ of decay functions.

## 4   The $\gamma$-prophet inequality

We study in this section the $\gamma$-prophet inequality, where all the decay functions are equal to $x \mapsto \gamma x$, for some $\gamma \in [0, 1]$. For any algorithm $\mathsf{ALG}$ with stopping time $\tau$ and random variables $X_1, \ldots, X_n$, if the observation order is $\pi$, we use the notation

$$\mathsf{ALG}^\gamma(X_1, \ldots, X_n) = \max\{X_{\pi(\tau)}, \gamma X_{\pi(\tau-1)}, \ldots, \gamma X_{\pi(1)}\} .$$

and we denote by $\mathsf{CR}^\gamma(\mathsf{ALG})$ the competitive ratio of $\mathsf{ALG}$ in this setting. In the following, we provide theoretical guarantees for the $\gamma$-prophet inequality.

For each observation order, we first derive upper bounds on the competitive ratio of any algorithm, depending on $\gamma$, using only hard instances satisfying the condition of Proposition 3.3. This would guarantee that the upper bounds extend to the $\mathcal{D}$-prophet inequality if $\gamma_\mathcal{D} = \gamma$. Then, we design single-threshold algorithms with well-chosen thresholds depending on $\gamma$ and the distributions, with competitive ratios improving with $\gamma$. A crucial property of single-threshold algorithms, which we use to estimate their competitive ratios, is that their reward satisfies

$$\mathsf{ALG}^\gamma(X_1, \ldots, X_n) = \mathsf{ALG}^0(X_1, \ldots, X_n) + \gamma(\max_i X_i)\mathbb{1}_{(\max_i X_i < \theta)} . \tag{4}$$

The additional term appearing due to $\gamma$ depends only on $\max_{i \in [n]} X_i$, which is the reward of the prophet against whom we compete. This property is not satisfied by more general class of algorithms such as multiple-threshold algorithms, where each observation $X_{\pi(i)}$ is compared with a threshold $\theta_i$.

**Remark 4.1.** *We only consider instances with continuous distributions in the proofs of lower bounds. The thresholds $\theta$ considered are such that $\Pr(\max_{i \in [n]} X_i \geq \theta) = g(\gamma, n, \pi)$, with $g$ depending on $\gamma$, the order model $\pi$ and the size of the instance $n$. Such a threshold is always guaranteed to exist when the distributions are continuous. However, as in the prophet inequality, the algorithms can be easily adapted to non-continuous distributions by allowing stochastic tie-breaking. A detailed strategy for doing this can be found for example in [Correa et al., 2021c].*

Before delving into the study of the different models, we provide generic lower and upper bounds, which depend solely on the bounds of the classical prophet inequality and $\gamma$.

**Proposition 4.2.** *In any order model, if $\alpha$ is a lower bound in the classical prophet inequality, and $\beta$ an upper bound, then, in the $\gamma$-prophet inequality*

> *(i) there exists a trivial algorithm with a competitive ratio of at least $\max\{\gamma, \alpha\}$,*
>
> *(ii) the competitive ratio of any algorithm is at most $(1 - \gamma)\beta + \gamma$.*

## 4.1 Adversarial order

We first consider the adversarial order model, and prove the upper bound of $\frac{1}{2-\gamma}$. Then, we provide a single-threshold algorithm with a competitive ratio matching this upper bound, hence fully solving the $\gamma$-prophet inequality in this adversarial order model.

**Theorem 4.3.** *In the adversarial order model, the competitive ratio of any algorithm is at most $\frac{1}{2-\gamma}$. Furthermore, there exists a single threshold algorithm with a competitive ratio $\frac{1}{2-\gamma}$: given any instance $(F_1, \ldots, F_n)$, this is achieved with the threshold $\theta$ satisfying*

$$\Pr_{X_1 \sim F_1, \ldots, X_n \sim F_n}(\max_{i \in [n]} X_i \leq \theta) = \frac{1}{2-\gamma} \ .$$

The upper bound in the previous theorem is proved using instances satisfying the condition of Proposition 3.3. Hence it extends to the $D_\infty$- then to the $\mathcal{D}$-prophet inequality, with $\gamma = \gamma_{\mathcal{D}}$, by Proposition 3.3 and Theorem 2.1.

## 4.2 Random order

Consider now that the items are observed in a uniformly random order $X_{\pi(1)}, \ldots, X_{\pi(n)}$, and $X^* = \max_{i \in [n]} X_i$. As for the adversarial model, we first prove an upper bound on the competitive ratio as a function of $\gamma$, and then prove a lower bound for a single-threshold algorithm. However, for this model, there is a gap between the two bounds, as illustrated in Figure 1.

We first prove an upper bound that depends on $\gamma$, matching the upper bound $\sqrt{3} - 1$ of Correa et al. [2021c] when $\gamma = 0$ and equal to 1 when $\gamma = 1$. Our single-threshold algorithm has a competitive ratio of at least $(1 - \frac{1}{e})$ when $\gamma = 0$, which is the best competitive ratio of a single threshold algorithm in the prophet inequality [Esfandiari et al., 2017, Correa et al., 2021c], and equal to 1 for $\gamma = 1$.

**Theorem 4.4.** *The competitive ratio of any algorithm $\mathsf{ALG}$ in the $\gamma$-prophet inequality with random order satisfies*
$$CR^\gamma(\mathsf{ALG}) \leq (1 - \gamma)^{3/2}(\sqrt{3 - \gamma} - \sqrt{1 - \gamma}) + \gamma \ .$$

*Furthermore, denoting by $p_\gamma$ is the unique solution to the equation $1 - (1 - \gamma)p = \frac{1-p}{-\log p}$, the single-threshold algorithm $\mathsf{ALG}_\theta$ with $\Pr_{X_1 \sim F_1, \ldots, X_n \sim F_n}(\max_{i \in [n]} X_i \leq \theta) = p_\gamma$ satisfies*

$$CR^\gamma(\mathsf{ALG}) \geq 1 - (1 - \gamma)p_\gamma \ .$$

Similarly to the adversarial order model, we used instances satisfying the condition of Proposition 3.3 to prove the upper bound, thus it extends to the $\mathcal{D}$-prophet inequality with $\gamma = \gamma_{\mathcal{D}}$.

While the equation defining $p_\gamma$ cannot be solved analytically, the solution can easily be computed numerically for any $\gamma \in [0, 1]$. Before moving to the IID case, we propose in the following a more explicit lower bound derived from Theorem 4.4.

**Corollary 4.4.1.** *In the random order model, the single threshold algorithm with a threshold $\theta$ satisfying $\Pr(\max_{i \in [n]} X_i \geq \theta) = \frac{1/e}{1 - (1 - 1/e)\gamma}$ has a competitive ratio of at least $1 - \frac{(1-\gamma)/e}{1 - (1 - 1/e)\gamma}$.*

## 4.3 IID Random Variables

In the classical IID prophet inequality, [Hill and Kertz, 1982] showed that the competitive ratio of any algorithm is at most $\approx 0.745$. The proof of this upper bound is hard to generalize for the IID $\gamma$-prophet inequality. As an alternative, we prove a weaker upper bound, which equals $\approx 0.778$

for $\gamma = 0$ and 1 for $\gamma = 1$, and the proof relies on instances of arbitrarily large size satisfying the condition of Proposition 3.3, hence the upper bound can be extended to the $\mathcal{D}$-prophet inequality.

Subsequently, we present a single-threshold algorithm with the same competitive ratio as the random order algorithm. However, the proof is different, leveraging the fact that the variables are identically distributed. More precisely, we introduce a single-threshold algorithm with guarantees that depend on the size $n$ of the instance, then we show that its competitive ratio is at least that of the algorithm presented in Theorem 4.4, with equality when $n$ approaches infinity.

Although it might look surprising that the obtained competitive ratio in the IID model is not better than that of the random-order model, the same behavior occurs in the classical prophet inequality. Indeed, Li et al. [2022] established that no single-threshold algorithm can achieve a competitive ratio better than $1 - 1/e$ in the standard prophet inequality with IID random variables, which is also the best possible with a single-threshold algorithm in the random order. However, considering larger classes of algorithms, the competitive ratios achieved in the IID model are better than those of the random order model.

We describe the algorithm and give a first lower bound on its reward depending on the size of the instance in the following lemma.

**Lemma 4.5.** *Let $a_{n,\gamma}$ be the unique solution of the equation $\left( \frac{1}{(1-a/n)^n} - 1 \right) \left( \frac{1}{a} - 1 \right) = \gamma$, then for any IID instance $X_1, \ldots, X_n$, the algorithm with threshold $\theta$ satisfying $\Pr(X_1 > \theta) = \frac{a_{n,\gamma}}{n}$ has a reward of at least*

$$\frac{1}{a_{n,\gamma}} \left( 1 - \left( 1 - \frac{a_{n,\gamma}}{n} \right)^n \right) \mathbb{E}[\max_{i \in [n]} X_i] .$$

We can prove that the reward presented in the Lemma above is strictly better than that of the random order model. However, both are asymptotically equal as we show in the following theorem.

**Theorem 4.6.** *The competitive ratio of any algorithm in the IID $\gamma$-prophet inequality is at most*

$$U(\gamma) = 1 - (1 - \gamma) \frac{e^2 \log(3 - \gamma) - (2 - \gamma)}{2(2e^2 - 1) - (3e^2 - 1)\gamma} .$$

*In particular, $U$ is increasing, $U(0) = \frac{4 - \log 3}{2(2 - \frac{1}{e}^2)} \approx 0.778$ and $U(1) = 1$. Furthermore, there exists a single-threshold algorithm $\mathsf{ALG}_\theta$ satisfying*

$$CR^\gamma(\mathsf{ALG}_\theta) \geq 1 - (1 - \gamma)p_\gamma ,$$

*where $p_\gamma$ is defined in Theorem 4.4.*

To prove the upper bound, we used instances satisfying the condition of Proposition 3.3, guaranteeing that it remains true in the $D_\infty$-prophet inequality with $\gamma = \gamma_\mathcal{D}$. On the other hand, Theorem 2.1 ensures that the upper bound extends to the $\mathcal{D}$-prophet inequality, but with an additional $O(1/n^{1/3})$ term. The latter does change the result, as we considered instances of arbitrarily large size $n \to \infty$.

## 5 Conclusion

In this paper, we addressed the $\mathcal{D}$-prophet inequality problem, which models a very broad spectrum of online selection scenarios, accommodating various observation order models and allowing to revisit rejected candidates at a cost. The problem extends the classic prophet inequality, corresponding to the special case where all decay functions are zero. The main result of the paper is a reduction from the general $\mathcal{D}$-prophet inequality to the $\gamma$-prophet inequality, where all the decay functions equal to $x \mapsto \gamma x$ for some constant $\gamma \in [0, 1]$. Subsequently, we provide algorithms and upper bounds for the $\gamma$-prophet inequality, which remain valid, by the previous reduction, in the $\mathcal{D}$-prophet inequality. Notably, the proved upper and lower bounds match each other for the adversarial order model, hence completely solving the problem. Our analysis paves the way for more practical applications of prophet inequalities, and advances efforts towards closing the gap between theory and practice in online selection problems.

**Limitations and future work**

**Better upper bounds in the $D_\infty$-prophet inequality.** Proposition 3.3 establishes that upper bounds proved in the $\gamma$-prophet inequality using instances of random variables with support in some set $\{0, a, b\}$ remain true in the $D_\infty$-prophet inequality, hence in the $\mathcal{D}$-prophet inequality by Theorem 2.1. This is enough to establish a tight upper bound in the adversarial order model, but not in the random order and IID models. An interesting question to explore is if more general upper bounds can be extended, or not, from the $\gamma$- to the $\mathcal{D}$-prophet inequality.

**Algorithms for the $\gamma$-prophet inequality.** As explained in Section 4, our analysis of the competitive ratio of single-threshold algorithms relies on the identity (4), which is not satisfied for instance by multiple-threshold algorithms. In the adversarial order model, we proved that the optimal competitive ratio $1/(2 - \gamma)$ can be achieved with a single-threshold algorithm. However, this is not the case in the random order or IID models. An interesting research avenue is to study other classes of algorithms in the $\gamma$-prophet inequality.

## Acknowledgements

This research was supported in part by the French National Research Agency (ANR) in the framework of the PEPR IA FOUNDRY project (ANR-23-PEIA-0003) and through the grant DOOM ANR-23-CE23-0002. It was also funded by the European Union (ERC, Ocean, 101071601). Views and opinions expressed are however those of the author(s) only and do not necessarily reflect those of the European Union or the European Research Council Executive Agency. Neither the European Union nor the granting authority can be held responsible for them.

Dorian Baudry thanks the support of the French National Research Agency: ANR-19-CHIA-02 SCAI, ANR-22-SRSE-0009 Ocean, and ANR-23-CE23-0002 Doom. Dorian Baudry was partially funded by UK Research and Innovation (UKRI) under the UK government's Horizon Europe funding guarantee [grant number EP/Y028333/1].

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

# A   From $\mathcal{D}$-prophet to the $D_\infty$-prophet inequality

In this section, we prove the reduction from the $\mathcal{D}$-prophet to the $D_\infty$-prophet inequality problem in the adversarial and random order models, and the reduction up to an additional $O(n^{-1/3})$ term in the IID model. First, we prove Corollary 2.1.1, which is the principal implication of Theorem 2.1.

## A.1   Proof of Corollary 2.1.1

*Proof.* Let us denote $\mathsf{A}_{*,\infty}$ the algorithm taking optimal decisions for any instance in the $D_\infty$-prophet inequality (obtained via dynamic programming). Then, by Theorem 2.1 we obtain for the adversarial and random order models that

$$\sup_{\mathsf{ALG}} \mathsf{CR}^{\mathcal{D}}(\mathsf{ALG}) \leq \inf_I \sup_{\mathsf{A}} \frac{\mathbb{E}[\mathsf{A}^{D_\infty}(I)]}{\mathbb{E}[\mathsf{OPT}(I)]} = \inf_I \frac{\mathbb{E}[\mathsf{A}_{*,\infty}^{D_\infty}(I)]}{\mathbb{E}[\mathsf{OPT}(I)]} = \mathsf{CR}^{D_\infty}(\mathsf{A}_{*,\infty}) = \sup_{\mathsf{A}} \mathsf{CR}^{D_\infty}(\mathsf{A}) \ . \tag{5}$$

Since $\mathsf{CR}^{\mathcal{D}}(\mathsf{ALG}) \geq \mathsf{CR}^{D_\infty}(\mathsf{ALG})$ for any algorithm, we deduce that (5) is an equality. If we consider now any algorithm $\bar{\mathsf{A}}_\infty$ that is optimal for the $D_\infty$-prophet inequality, not necessarily $\mathsf{A}_{*,\infty}$, then Equation (5) provides

$$\mathsf{CR}^{\mathcal{D}}(\bar{\mathsf{A}}_\infty) \geq \mathsf{CR}^{D_\infty}(\bar{\mathsf{A}}_\infty) = \sup_{\mathsf{ALG}} \mathsf{CR}^{\mathcal{D}}(\mathsf{ALG}) \ .$$

The previous inequality is again an equality, and it implies that $\bar{\mathsf{A}}_\infty$ is also optimal, in the sense of the competitive ratio, for the $\mathcal{D}$-prophet inequality, and

$$\mathsf{CR}^{\mathcal{D}}(\bar{\mathsf{A}}_\infty) = \mathsf{CR}^{D_\infty}(\bar{\mathsf{A}}_\infty) \ .$$

$\square$

## A.2   Auxilary Lemma

The efficiency of the proof scheme introduced in Section 2.1 relies on the following key argument: if $(D_j)_{j \geq 1}$ converges pointwise to $D_\infty$, then for any algorithm $\mathsf{A}$ and any instance $I$, the output of $\mathsf{A}$ when all the decay functions are equal to $D_m$ converges to its output when all the decay functions are equal to $D_\infty$. If $X_1, \ldots, X_n$ are the realizations of $I$ observed by $\mathsf{A}$ and if $\sigma$ is the order in which they are observed, then denoting $\tau$ the stopping time of $\mathsf{A}$ we can write that

$$\mathbb{E}[\mathsf{A}^{D_m}(I)] - \mathbb{E}[\mathsf{A}^{D_\infty}(I)]$$
$$= \mathbb{E}[\max\{X_{\sigma(\tau)}, D_m(\max_{i<\tau} X_{\sigma(i)})\}] - \mathbb{E}[\max\{X_{\sigma(\tau)}, D_\infty(\max_{i<\tau} X_{\sigma(i)})\}]$$
$$\leq \max_{\substack{\pi \in \mathcal{S}_n \\ q \in [n]}} \left\{ \mathbb{E}[\max\{X_{\pi(q)}, D_m(\max_{i<q} X_{\pi(i)})\}] - \mathbb{E}[\max\{X_{\pi(q)}, D_\infty(\max_{i<q} X_{\pi(i)})\}] \right\} \ ,$$

where $\mathcal{S}_n$ is the set of all permutations of $[n]$. The latter upper bound is independent of $\sigma$ and $\mathsf{A}$. We show in the following lemma that it converges to 0 when $m \to \infty$.

**Lemma A.1.** *Let $\mathcal{S}_n$ be the set of all permutations of $[n]$. For any fixed instance $I = (F_1, \ldots, F_n)$, considering $X_i \sim F_i$ for all $i \in [n]$, define for all $m \geq 1$*

$$\epsilon_m(I) = \max_{\substack{\pi \in \mathcal{S}_n \\ q \in [n]}} \left\{ \mathbb{E}[\max\{X_{\pi(q)}, D_m(\max_{i<q} X_{\pi(i)})\}] - \mathbb{E}[\max\{X_{\pi(q)}, D_\infty(\max_{i<q} X_{\pi(i)})\}] \right\} \ .$$

*If $\mathbb{E}[\max_{i \in [n]} X_i] < \infty$, then $\lim_{m \to \infty} \epsilon_m(I) = 0$.*

*Proof.* Let us denote $f_1, \ldots, f_n$ the respective probability density functions of $X_1, \ldots, X_q$. For any $q \in [n]$ and $\pi \in \mathcal{S}_n$, let us define for all $m \geq 0$ the function $\varphi_m^{\pi,q} : [0,\infty)^q \to [0,\infty)$ by $\varphi_m^{\pi,q}(x_1, \ldots, x_q) = \max\{x_{\pi(q)}, D_m(\max_{i<q} x_{\pi(i)})\} - \max\{x_{\pi(q)}, D_\infty(\max_{i<q} x_{\pi(i)})\}$. $\varphi_m^{\pi,q}$ is positive because $D_m \geq D_\infty$. The sequence $(\varphi_m^{\pi,q})_m$ is non-increasing, converges to 0 pointwise, and is dominated by $(x_1, \ldots, x_q) \mapsto \max_{i \in [q]} x_i$, which is integrable with respect to the probability

measure $(x_1, \ldots, x_q) \mapsto \prod_{i=1}^q f(x_i)$. Therefore, using the dominated convergence theorem, we deduce that $\lim_{m \to \infty} \mathbb{E}[\varphi_m^{\pi,q}(X_1, \ldots, X_q)] = 0$. It follows that

$$\lim_{m \to \infty} \epsilon_m(I) = \lim_{m \to \infty} \left( \max_{\substack{\pi \in \mathcal{S}_n \\ q \in [n]}} \mathbb{E}[\varphi_m^{\pi,q}(X_1, \ldots, X_q)] \right) = 0 \ .$$

$\square$

### A.3 Proof of Theorem 2.1

*Proof of Theorem 2.1.* We provide a separate proof for each of the adversarial order, random order and IID models.

**Adversarial order**  Let $I = (F_1, \ldots, F_n)$ be any instance and $X_i \sim F_i$ for all $i \in [n]$. Consider the instance $I_m = (Y_1, \ldots, Y_{mn})$, where $Y_{km} \sim F_k$ for any $k \in [n]$ and $Y_i = 0$ a.s. for all $i \notin \{m, 2m, \ldots, mn\}$. It is clear that no reasonable algorithm would stop at a zero value: if the current observation is 0 it is preferable to wait for a non-null value, or it would have been preferable to stop at the previous non-null value. Hence, $\tau$ is a multiple of $m$: $\tau = \rho m$ for some $\rho \in \mathbb{N}^\star$. Given that $D_j(0) = 0$ for all $j$ and the sequence $(D_j)_j$ is non-increasing, we have that

$$\begin{aligned}
\mathbb{E}[\mathsf{ALG}^{\mathcal{D}}(I_m)] &= \mathbb{E}[\max_{i \leq \tau} D_{\tau-i}(Y_i)] \\
&= \mathbb{E}[\max_{k \leq \rho} D_{\tau-km}(Y_{km})] \\
&= \mathbb{E}[\max_{k \leq \rho} D_{\rho m - km}(X_k)] \\
&= \mathbb{E}[\max\{X_\rho, \max_{k < \rho} D_{(\rho-k)m}(X_k)\}] \\
&\leq \mathbb{E}[\max\{X_\rho, \max_{k < \rho} D_m(X_k)\}] \\
&\leq \mathbb{E}[\max\{X_\rho, \max_{k < \rho} D_\infty(X_k)\}] + \epsilon_m(I) \ ,
\end{aligned}$$

where $\epsilon_m(I)$ is defined in Lemma A.1. We can then use that the first right-hand term is the output of some other algorithm that would choose a stopping time $\rho$ when facing $I$ in the context of the $D_\infty$-prophet inequality. More precisely, consider the algorithm $\mathsf{A}_m$ which, given any instance $I = (F_1, \ldots, F_n)$, simulates the behavior of $\mathsf{ALG}$ facing the sequence $I_m$, where at each step $i \in [mn]$

- if $i \notin \{m, \ldots, nm\}$: $\mathsf{ALG}$ observes $Y_i = 0$,

- otherwise, if $i = km$ for some $k \in [n]$: $\mathsf{A}_m$ observes $X_k$ and $\mathsf{ALG}$ observes $Y_{km} = X_k$

- if $\mathsf{ALG}$ stops on $Y_{\rho m}$, then $\mathsf{A}_m$ also stops, and its reward is $X_\rho$.

$\mathsf{A}_m(X_1, \ldots, X_n)$ stops at the same value as $\mathsf{ALG}(Y_1, \ldots, Y_m)$, their reward in the $D_\infty$-prophet inequality is the same, and since $\max_{i \in [n]} X_i = \max_{i \in [mn]} Y_i$ this yields to

$$\mathsf{CR}^{\mathcal{D}}(\mathsf{ALG}) \leq \frac{\mathbb{E}[\mathsf{ALG}^{\mathcal{D}}(I_m)]}{\mathbb{E}[\mathsf{OPT}(I_m)]} \leq \frac{\mathbb{E}[\mathsf{A}_m^{D_\infty}(I)] + \epsilon_m(I)}{\mathbb{E}[\mathsf{OPT}(I)]} \leq \sup_{\mathsf{A: \ algo}} \frac{\mathbb{E}[\mathsf{A}^{D_\infty}(I)]}{\mathbb{E}[\mathsf{OPT}(I)]} + \frac{\epsilon_m(I)}{\mathbb{E}[\mathsf{OPT}(I)]} \ ,$$

and taking the limit when $m \to \infty$ gives the result, by making the second term vanish.

**Random order**  Let $I = (F_1, \ldots, F_n)$ be an instance of distributions and $X_i \sim F_i$ for $i \in [n]$. Using the notation $\delta_0$ for the Dirac distribution in 0, we consider $I_m = (F_1, \ldots, F_n, \delta_0, \ldots, \delta_0)$ containing $m$ copies of $\delta_0$ so that the observations from this instance always contain at least $m$ null values. Let $Y_1, \ldots, Y_m$ be a realization of this instance. For simplicity, say that $Y_i = X_i$ for $i \in [n]$, and $Y_i = 0$ for $i > n$.

We first show that when $m \to \infty$, since the observation order is drawn uniformly at random, the algorithm observes a large number of zeros between every two random variables drawn from $(F_1, \ldots, F_n)$. Let us denote by $\pi$ the uniformly random order in which the observations are received,

i.e. the algorithms observes $Y_{\pi(1)}, Y_{\pi(2)}, \ldots$, and let $\ell \geq 1$ be some positive integer, and $t_1, \ldots, t_n$ be the increasing indices in which the variables $Y_1, \ldots, Y_n$ are observed, i.e. $t_1 < \ldots < t_n$ and $\{t_1, \ldots t_n\} = \{\pi^{-1}(1), \ldots, \pi^{-1}(n)\}$. Therefore, any observation outside $\{Y_{\pi(t_1)}, \ldots, Y_{\pi(t_1)}\}$ is zero. Using the notation $L = \min_{i \in [n-1]} |t_{i+1} - t_i|$, we obtain that

$$
\begin{aligned}
\Pr(L \leq \ell) &= \Pr(\cup_{i=1}^{n-1} \{t_{i+1} - t_i \leq \ell\}) \\
&= \Pr\left(\cup_{k=1}^{n} \cup_{j=1}^{k-1} \left\{|\pi^{-1}(k) - \pi^{-1}(j)| \leq \ell\right\}\right) \\
&\leq \frac{n(n-1)}{2} \Pr(|\pi^{-1}(1) - \pi^{-1}(2)| \leq \ell) \\
&= \frac{n(n-1)}{2} \Pr\left(\left(\cup_{k=1}^{n+m} \left(\pi^{-1}(1) = k, \pi^{-1}(2) \in \{k - \ell, \ldots, k + \ell\} \setminus \{k\}\right)\right) \right. \\
&\leq \frac{n(n-1)}{2} \times (n+m) \times \frac{1}{n+m} \times \frac{2\ell}{n+m-1} \\
&\leq \frac{n^2 \ell}{m} \ .
\end{aligned}
$$

Taking $\ell = \sqrt{m}$, we find that $\Pr(L \leq \ell) \leq n^2/\sqrt{m}$. Therefore, for any algorithm ALG, observing that the reward of $\mathsf{ALG}^D$ is at most $\max_{i \in [n]} X_i$ a.s., and by independence of $\max_{i \in [n]} X_i$ and $L$, we deduce that

$$
\begin{aligned}
\mathbb{E}[\mathsf{ALG}^{\mathcal{D}}(I_m)] &= \mathbb{E}[\mathsf{ALG}^{\mathcal{D}}(I_m) \mathbb{1}_{L > \ell}] + \mathbb{E}[\mathsf{ALG}^{\mathcal{D}}(I_m) \mathbb{1}_{L \leq \ell}] \\
&\leq \mathbb{E}[\mathsf{ALG}^{\mathcal{D}}(I_m) \mid L > \ell] + \mathbb{E}[(\max_{i \in [n]} X_i) \mathbb{1}_{L \leq \ell}] \\
&\leq \mathbb{E}[\mathsf{ALG}^{\mathcal{D}}(I_m) \mid L > \ell] + \mathbb{E}[\max_{i \in [n]} X_i] \frac{n^2}{\sqrt{m}} \ .
\end{aligned}
\tag{6}
$$

Let us denote $\tau$ the stopping time of ALG and $t_\rho = \max_{j \in [n]} \{t_j : t_j \leq \tau\}$ the last time when a variable $(X_j)_{j \in [n]}$ was observed by ALG. The sequence of functions $(D_j)_{j \geq 1}$ is non-increasing, hence

$$
\begin{aligned}
\mathbb{E}[\mathsf{ALG}^{\mathcal{D}}(I_m) \mid L > \ell] &= \mathbb{E}[\max_{i \in [\tau]} D_{\tau-i}(Y_{\pi(i)}) \mid L > \ell] \\
&= \mathbb{E}[\max_{j \leq i} D_{\tau-t_j}(Y_{\pi(t_j)}) \mid L > \ell] \tag{7} \\
&\leq \mathbb{E}[\max_{j \leq i} D_{t_\rho-t_j}(Y_{\pi(t_j)}) \mid L > \ell] \tag{8} \\
&= \mathbb{E}[\max \left\{Y_{\pi(t_\rho)}, \max_{j < \rho} D_{t_\rho-t_j}(Y_{\pi(t_j)})\right\} \mid L > \ell] \\
&\leq \mathbb{E}[\max \left\{Y_{\pi(t_\rho)}, \max_{j < \rho} D_\ell(Y_{\pi(t_j)})\right\}] \ , \tag{9}
\end{aligned}
$$

Equation (7) holds because the only non-zero values up to step $\tau$ are $(Y_{\pi(t_j)})_{j \in [\rho]}$. Inequality (8) uses that the sequence $(D_j)_{j \geq 1}$ is non-increasing, and (9) uses, in addition to that, the independence of $L$ and $(Y_{\pi(t_j)})_{j \in [n]}$. We now argue that the term $\mathbb{E}[\max \left\{Y_{\pi(t_\rho)}, \max_{j < \rho} D_\ell(Y_{\pi(t_j)})\right\}]$ is the expected reward of an algorithm in the $D_\ell$-inequality. Given that $\pi$ is a uniform random permutation of $[n+m]$ and by definition of $t_1, \ldots, t_n$, the application $\sigma : k \in [n] \mapsto \pi(t_k)$ is a random permutation of $[n]$. Therefore we consider the algorithm $\mathsf{A}_m$ that receives as input the instance $I = (F_1, \ldots, F_n)$, then considers the array $u = (1, \ldots, 1, 0, \ldots, 0)$ composed of $n$ values equal to $1$ and $m$ zero values, and a uniformly random permutation $\pi$ of $[n+m]$, then simulates $\mathsf{ALG}^D(I_m)$ as follows: at each step $j \in [n+m]$

- if $u_{\pi(j)} = 0$, then ALG observes the value $Y_{\pi(j)} = 0$,

- if $u_{\pi(j)} = 1$, then $\mathsf{A}_m$ observes the next value $X_{\sigma(k)}$, and ALG observes $Y_{\pi(j)} = X_{\sigma(k)}$,

- when ALG decides to stop, $\mathsf{A}_m$ also stops, and its reward is the current value $X_{\sigma(k)}$.

With this construction, $(Y_j)_{j \in [n+m]}$ is indeed a realization of the instance $I_m$, and $\mathsf{A}_m$ stops on the last value sampled from $F_1, \ldots, F_n$ observed by ALG. Therefore, denoting $\rho$ the stopping time of

$A_m$, and $\epsilon_\ell(I)$ as defined in Lemma A.1, we have

$$\mathbb{E}[\mathsf{ALG}^{\mathcal{D}}(I_m) \mid L > \ell] \leq \mathbb{E}[\max \{Y_{\pi(t_\rho)}, \max_{j < \rho} D_\ell(Y_{\pi(t_j)})\}]$$

$$= \mathbb{E}[\max \{X_{\sigma(\rho)}, \max_{j < \rho} D_\ell(X_{\sigma(t_j)})\}]$$

$$= \mathbb{E}[\mathsf{A}_m^{D_\ell}(I)]$$

$$\leq \mathbb{E}[\mathsf{A}_m^{D_\infty}(I)] + \epsilon_\ell(I)$$

$$\leq \sup_{\mathsf{A}:\mathsf{algo}} \mathbb{E}[\mathsf{A}^{D_\infty}(I)] + \epsilon_\ell(I) .$$

Taking $\ell = \sqrt{m}$ and substituting into Equation (6), then observing that $\mathbb{E}[\mathsf{OPT}(I)] = \mathbb{E}[\mathsf{OPT}(I_m)]$, gives that

$$\mathsf{CR}^{\mathcal{D}}(\mathsf{ALG}) \leq \frac{\mathbb{E}[\mathsf{ALG}^{\mathcal{D}}(I_m)]}{\mathbb{E}[\mathsf{OPT}(I_m)]} \leq \sup_{\mathsf{A}:\mathsf{algo}} \frac{\mathbb{E}[\mathsf{A}^{D_\infty}(I)]}{\mathbb{E}[\mathsf{OPT}(I)]} + \frac{\epsilon_{\sqrt{m}}(I)}{\mathbb{E}[\mathsf{OPT}(I)]} + \frac{n^2}{\sqrt{m}} .$$

Finally, taking $m \to \infty$ and using Lemma A.1, we deduce that

$$\mathsf{CR}^{\mathcal{D}}(\mathsf{ALG}) \leq \sup_{\mathsf{A}:\mathsf{algo}} \frac{\mathbb{E}[\mathsf{A}^{D_\infty}(I)]}{\mathbb{E}[\mathsf{OPT}(I)]} ,$$

which completes the proof for the random order.

**IID random variables**   For any probability distribution $F$ on $[0, \infty)$ and for any $n \geq 1$ we denote $\mathbb{E}[\mathsf{OPT}(F, n)]$ the expected maximum of $n$ independent random variables drawn from $F$, and for any algorithm $\mathsf{ALG}$ we denote $\mathbb{E}[\mathsf{ALG}^{\mathcal{D}}(F, n)]$ its expected output when given $n$ IID variable sampled from $F$ as input. The proof of Theorem 2.1 for this last model is much more technical than for previous models, so we first prove several auxiliary results that we will later use to provide a concise proof of the last part of the theorem.

**Lemma A.2.** *For any probability distribution $F$ and $n \geq 1, \Delta \geq 0$, we have*

$$\mathbb{E}[\textit{OPT}(F, n + \Delta)] \leq \left(1 + \frac{\Delta}{n}\right) \mathbb{E}[\textit{OPT}(F, n)] .$$

*Proof.* We first write

$$\Pr(\mathsf{OPT}(F, n + \Delta) > x) = 1 - F(x)^{n+\Delta}$$

$$= \left(1 + F(x)^n \frac{1 - F(x)^\Delta}{1 - F(x)^n}\right)(1 - F(x)^n)$$

$$= \left(1 + F(x)^n \frac{1 - F(x)^\Delta}{1 - F(x)^n}\right) \Pr(\mathsf{OPT}(F, n) > x) ,$$

and then use that

$$F(x)^\Delta = e^{\Delta \log(F(x))} \geq 1 + \Delta \log(F(x)) = 1 - \frac{\Delta}{n} \log(1/F(x)^n) \geq 1 - \frac{\Delta}{n} \left(\frac{1 - F(x)^n}{F(x)^n}\right) ,$$

so we directly obtain

$$F(x)^n \frac{1 - F(x)^\Delta}{1 - F(x)^n} \leq \frac{\Delta}{n} ,$$

which gives that

$$\Pr(\mathsf{OPT}(F, n + \Delta) > x) \leq \left(1 + \frac{\Delta}{n}\right) \Pr(\mathsf{OPT}(F, n) > x) .$$

As we consider non-negative random variables, it follows directly that

$$\mathbb{E}[\mathsf{OPT}(F, n + \Delta)] \leq \left(1 + \frac{\Delta}{n}\right) \mathbb{E}[\mathsf{OPT}(F, n)] .$$

$\square$

**Lemma A.3.** *Let $N \sim \mathcal{B}(m, \varepsilon)$ and let $n := \mathbb{E}[N] = \varepsilon m$, then we have*

$$\mathbb{E}[OPT(F, N)\mathbb{1}_{N \geq n+n^{2/3}}] \leq \frac{6}{n^{1/3}}\mathbb{E}[OPT(F, n + n^{2/3})] ,$$

$$\mathbb{E}[OPT(F, N)] \leq \left(1 + \frac{3}{n^{1/3}}\right)\mathbb{E}[OPT(F, n + n^{2/3})] .$$

*Proof.* Let $\Delta, s > 0$ such that $\Delta \leq s$. For any $k \geq 1$ let $W_k = [s + (k-1)\Delta, s + k\Delta)$. $(W_k)_{k \geq 1}$ is a partition of $[s, \infty)$, thus we have

$$\mathbb{E}[\mathsf{OPT}(F, N)\mathbb{1}_{N \geq s}] = \sum_{k=1}^{\infty}\mathbb{E}[\mathsf{OPT}(F, N)\mathbb{1}_{N \in W_k}]$$

$$\leq \sum_{k=1}^{\infty}\mathbb{E}[\mathsf{OPT}(F, s + k\Delta)\mathbb{1}_{N \in W_k}]$$

$$= \sum_{k=1}^{\infty}\mathbb{E}[\mathsf{OPT}(F, s + k\Delta)]\Pr(N \in W_k)$$

$$\leq \sum_{k=1}^{\infty}\left(1 + \frac{k\Delta}{s}\right)\mathbb{E}[\mathsf{OPT}(F, s)]\Pr(N \in W_k)$$

$$= \left(\Pr(N \geq s) + \frac{\Delta}{s}\sum_{k=1}^{\infty}k\Pr(N \in W_k)\right)\mathbb{E}[\mathsf{OPT}(F, s)] ,$$

where we used Lemma A.2 in the penultimate inequality. Furthermore, observing that

$$\sum_{k=1}^{\infty}k\Pr(N \in W_k) = \sum_{k=1}^{\infty}\sum_{\ell=0}^{k-1}\Pr(N \in W_k) = \sum_{\ell=0}^{\infty}\sum_{k=\ell+1}^{\infty}\Pr(N \in W_k) = \sum_{\ell=0}^{\infty}\Pr(N \geq s + \ell\Delta) ,$$

we obtain, given $\Delta \leq s$, that

$$\mathbb{E}[\mathsf{OPT}(F, N)\mathbb{1}_{N \geq s}] \leq \left(\Pr(N \geq s) + \frac{\Delta}{s}\sum_{k=0}^{\infty}\Pr(N \geq s + k\Delta)\right)\mathbb{E}[\mathsf{OPT}(F, s)] \qquad (10)$$

$$\leq \left(2\sum_{k=0}^{\infty}\Pr(N \geq s + k\Delta)\right) . \qquad (11)$$

$N$ is a Binomial random variable with expectation $n$. Therefore, Chernoff's inequality gives for any $\delta \geq 0$ that

$$\Pr(N \geq (1 + \delta)n) \leq \exp\left(-\frac{\delta^2 n}{2 + \delta}\right) \leq \exp\left(-\frac{\min(\delta, \delta^2)n}{3}\right) ,$$

where the second inequality can be derived by treating separately $\delta < 1$ and $\delta \geq 1$. In particular, for any $k \geq 1$, taking $\delta = \frac{k\Delta}{n}$ such that $\Delta \leq n$ yields

$$\Pr(N \geq n+k\Delta) \leq \exp\left(-\frac{\min(k\Delta, k^2\Delta^2/n)}{3}\right) \leq \exp\left(-\frac{k\min(\Delta, \Delta^2/n)}{3}\right) = \exp\left(-\frac{k\Delta^2}{3n}\right) .$$

Substituting this Inequality into (11) with $s = n + \Delta$, we obtain

$$\mathbb{E}[\mathsf{OPT}(F, N)\mathbb{1}_{N \geq n+\Delta}] \leq \left(2\sum_{k=1}^{\infty}\Pr(N \geq n + k\Delta)\right)\mathbb{E}[\mathsf{OPT}(F, n + \Delta)]$$

$$\leq \left(2\sum_{k=1}^{\infty}\exp\left(-\frac{k\Delta^2}{3n}\right)\right)\mathbb{E}[\mathsf{OPT}(F, n + \Delta)]$$

$$= \frac{2}{\exp\left(\frac{\Delta^2}{3n}\right) - 1}\mathbb{E}[\mathsf{OPT}(F, n + \Delta)]$$

$$\leq \frac{6n}{\Delta^2}\mathbb{E}[\mathsf{OPT}(F, n + \Delta)] ,$$

and taking $\Delta = n^{2/3}$ proves the first inequality of the lemma.

Let us move now to the second inequality. We have
$$\mathbb{E}[\text{OPT}(F, N)\mathbb{1}_{N<s}] \leq \mathbb{E}[\text{OPT}(F, s)\mathbb{1}_{N<s}] = \mathbb{E}[\text{OPT}(F, s)]\Pr(N < s) ,$$
and thus, using Inequality (10), again with $s = n + \Delta$ and $\Delta = n^{2/3}$, it follows that
$$\mathbb{E}[\text{OPT}(F, N)] = \mathbb{E}[\text{OPT}(F, N)\mathbb{1}_{N<s}] + \mathbb{E}[\text{OPT}(F, N)\mathbb{1}_{N\geq s}]$$

$$\leq \left(1 + \frac{\Delta}{s} \sum_{k=0}^{\infty} \Pr(N \geq s + k\Delta)\right) \mathbb{E}[\text{OPT}(F, s)]$$

$$\leq \left(1 + \sum_{k=1}^{\infty} \Pr(N \geq n + k\Delta)\right) \mathbb{E}[\text{OPT}(F, s)]$$

$$\leq \left(1 + \frac{3}{n^{1/3}}\right) \mathbb{E}[\text{OPT}(F, n + \Delta)] .$$

$\square$

**Lemma A.4.** *Let* $\delta_1, \ldots, \delta_m \overset{iid}{\sim} \mathcal{B}(\varepsilon)$, *and* $N = \sum_{i=1}^{m} \delta_i$. *Denoting by* $n = \mathbb{E}[N] = \varepsilon m$, *if* $n \geq 4$ *then*
$$\mathbb{E}[N^2 OPT(F, N)] \leq \left(1 + \frac{8}{n^3}\right) n^2 \mathbb{E}[OPT(F, n + n^{2/3})] .$$

*Proof.* For all $k \in [m]$, denote by $N_k = \sum_{i=k}^{m} \delta_i$. We have that
$$N^2\text{OPT}(F, N) = \left(\sum_{i=1}^{m} \delta_i\right)^2 \text{OPT}(F, N)$$

$$= \left(\sum_{i=1}^{m} \delta_i^2 + 2\sum_{i<j} \delta_i\delta_j\right) \text{OPT}(F, N) ,$$

and observing that $\delta_i^2 = \delta_i$ for all $i$, we obtain in expectation
$$\mathbb{E}[N^2\text{OPT}(F, N)] = m\mathbb{E}[\delta_1\text{OPT}(F, \delta_1 + N_2)] + m(m - 1)\mathbb{E}[\delta_1\delta_2\text{OPT}(\delta_1 + \delta_2 + N_3)]$$
$$\leq m\mathbb{E}[\delta_1\text{OPT}(F, 1 + N_2)] + m(m - 1)\mathbb{E}[\delta_1\delta_2\text{OPT}(2 + N_3)]$$
$$= m\varepsilon\mathbb{E}[\text{OPT}(F, 1 + N_2)] + m(m - 1)\varepsilon^2\mathbb{E}[\text{OPT}(2 + N_3)]$$
$$= m\varepsilon\mathbb{E}[\text{OPT}(F, 1 + N_2)] + m^2\varepsilon^2\mathbb{E}[\text{OPT}(2 + N_3)] . \tag{12}$$
For $j \in \{1, 2\}$, the proof of Lemma A.3 can be easily adjusted to prove an upper bound on $\mathbb{E}[\text{OPT}(F, j + N_{j+1})]$, by first bounding $\mathbb{E}[\text{OPT}(F, j + N_{j+1})\mathbb{1}_{N_{j+1}\geq s}]$ then $\mathbb{E}[\text{OPT}(F, j + N_{j+1})\mathbb{1}_{N_{j+1}<s}]$. The concentration arguments remain the same, replacing $m$ by $m - j$. The expectation of $N_{j+1}$ is $\varepsilon(m - j) = n - \varepsilon j$, hence we obtain

$$\mathbb{E}[\text{OPT}(F, j + N_{j+1})] \leq \left(1 + \frac{3}{(n - \varepsilon j)^{1/3}}\right) \mathbb{E}[\text{OPT}(F, j + (n - \varepsilon j) + (n - \varepsilon j)^{2/3})]$$

$$= \left(1 + \frac{3}{(n - \varepsilon j)^{1/3}}\right) \mathbb{E}[\text{OPT}(F, j + n + n^{2/3})]$$

$$\leq \left(1 + \frac{3}{(n - 2)^{1/3}}\right) \mathbb{E}[\text{OPT}(F, 2 + n + n^{2/3})]$$

$$\leq \left(1 + \frac{4}{n^{1/3}}\right) \mathbb{E}[\text{OPT}(F, 2 + n + n^{2/3})] ,$$

where we used respectively in the last inequalities that $j \leq 2$ and $n \geq 4$. Furthermore, Lemma A.2 gives that

$$\mathbb{E}[\text{OPT}(F, j + N_{j+1})] \leq \left(1 + \frac{4}{n^{1/3}}\right)\left(1 + \frac{2}{n + n^{2/3}}\right) \mathbb{E}[\text{OPT}(F, n + n^{2/3})]$$

$$\leq \left(1 + \frac{6}{n^{1/3}}\right) \mathbb{E}[\text{OPT}(F, n + n^{2/3})] ,$$

where the last inequality is true for $n \geq 4$. Finally, substituting into 12 yields

$$
\begin{aligned}
\mathbb{E}[N^2\mathsf{OPT}(F, N)] &\leq (m^2\varepsilon^2 + m\varepsilon)\left(1 + \frac{6}{n^{1/3}}\right)\mathbb{E}[\mathsf{OPT}(F, n + n^{2/3})] \\
&= (n^2 + n)\left(1 + \frac{6}{n^{1/3}}\right)\mathbb{E}[\mathsf{OPT}(F, n + n^{2/3})] \\
&= \left(1 + \frac{1}{n}\right)\left(1 + \frac{6}{n^{1/3}}\right)n^2\mathbb{E}[\mathsf{OPT}(F, n + n^{2/3})] \\
&\leq \left(1 + \frac{8}{n^{1/3}}\right)n^2\mathbb{E}[\mathsf{OPT}(F, n + n^{2/3})] \, .
\end{aligned}
$$

$\square$

**Lemma A.5.** *Let* $\delta_1, \ldots, \delta_m \overset{iid}{\sim} \mathcal{B}(\varepsilon)$, $N = \sum_{i=1}^m \delta_i$, $n = \mathbb{E}[N] = \varepsilon m$ *and* $L = \min_{i \neq j}\{|i - j| : \delta_i = 1, \delta_j = 1\}$, *then for any* $\ell \geq 0$ *we have*

$$
\mathbb{E}[\mathsf{OPT}(F, N)\mathbb{1}_{L \leq \ell}] \leq 7m\ell\varepsilon^2 \mathbb{E}[\mathsf{OPT}(F, n + n^{2/3})] \, .
$$

*Proof.* The random variables $N$ and $L$ are not independent, thus we need to adequately compute the distribution of $L$ conditional to $N$. For any $\ell \geq 0$ and $k \geq 2$ we have

$$
\begin{aligned}
\Pr(L \leq \ell, N = s) &= \Pr\left(L \leq \ell, \sum_{i=1}^m \delta_k = s\right) \\
&= \Pr\left(\cup_{i=1}^m \cup_{j=\max(1, i-\ell)}^{i-1}\left(\delta_i = \delta_j = 1, \sum_{i=1}^m \delta_k = s\right)\right) \\
&\leq m\ell\Pr\left(\delta_1 = \delta_2 = 1, \sum_{i=3}^m \delta_k = s - 2\right) \\
&= m\ell\binom{m-2}{s-2}\varepsilon^s(1 - \varepsilon)^{m-s+2},
\end{aligned}
$$

therefore

$$
\begin{aligned}
\Pr(L \leq \ell \mid N = s) &= \frac{\Pr(L \leq \ell, N = s)}{\Pr(N = s)} \\
&\leq \frac{m\ell\binom{m-2}{s-2}\varepsilon^s(1 - \varepsilon)^{m-s+2}}{\binom{m}{s}\varepsilon^s(1 - \varepsilon)^{m-s}} \\
&\leq m\ell\frac{\binom{m-2}{s-2}}{\binom{m}{s}} \\
&= m\ell\frac{s(s - 1)}{m(m - 1)} \\
&\leq \frac{\ell s^2}{m} \, .
\end{aligned}
$$

Using this inequality and Lemma A.4, we deduce that

$$
\begin{aligned}
\mathbb{E}[\mathsf{OPT}(F, N)\mathbb{1}_{L \leq \ell}] &= \mathbb{E}[\mathsf{OPT}(F, N)\Pr(L \leq \ell \mid N, \mathsf{OPT}(F, N))] \\
&= \mathbb{E}[\mathsf{OPT}(F, N)\Pr(L \leq \ell \mid N)] \\
&\leq \frac{\ell}{m}\mathbb{E}[N^2\mathsf{OPT}(F, N)] \\
&\leq \left(1 + \frac{8}{n^{1/3}}\right)\frac{\ell n^2}{m}\mathbb{E}[\mathsf{OPT}(F, n + n^{2/3})] \\
&= \left(1 + \frac{8}{n^{1/3}}\right)m\ell\varepsilon^2 \mathbb{E}[\mathsf{OPT}(F, n + n^{2/3})] \\
&= 7m\ell\varepsilon^2 \mathbb{E}[\mathsf{OPT}(F, n + n^{2/3})] \, .
\end{aligned}
$$

where we used that $1 + \frac{8}{n^{1/3}} \leq 7$ for $n \geq 4$.

$\square$

Using the previous lemmas, we can now prove the theorem. Let $m > n \geq 1$, $\Delta = n^{2/3}$, $\varepsilon = n/m$ and let $Q$ be the probability distribution of a random variable that is equal to $0$ with probability $1 - \varepsilon$, and drawn from $F$ with probability $\varepsilon$.

Let us consider $m$ i.i.d. variables $Y_1, \ldots, Y_m \sim Q$, and for each $i \in [m]$ we denote by $\delta_i$ the indicator that $Y_i$ is drawn from $F$. Define $N = \sum_{i=1}^{m} \delta_i \sim \mathcal{B}(m, \varepsilon)$ the number of random variables $Y_i$ drawn from the distribution $F$. In the following, we upper bound the competitive ratio of any algorithm by analyzing its ratio on this particular instance. For this, we first provide a lower bound on $\mathbb{E}[\mathsf{OPT}(Q, m)]$ using Lemma A.2, and obtain

$$\mathbb{E}[\mathsf{OPT}(F, n - \Delta)] \geq \frac{1}{1 + \frac{2\Delta}{n}} \mathbb{E}[\mathsf{OPT}(F, n + \Delta)] \geq \left(1 - \frac{2\Delta}{n}\right) \mathbb{E}[\mathsf{OPT}(F, n + \Delta)],$$

thus we have

$$
\begin{aligned}
\mathbb{E}[\mathsf{OPT}(Q, m)] &= \mathbb{E}[\mathsf{OPT}(F, N)] \\
&\geq \mathbb{E}[\mathsf{OPT}(F, N)\mathbb{1}_{N \geq n - \Delta}] \\
&\geq \mathbb{E}[\mathsf{OPT}(F, n - \Delta)\mathbb{1}_{N \geq n - \Delta}] \\
&= \mathbb{E}[\mathsf{OPT}(F, n - \Delta)] \Pr(N \geq n - \Delta) \\
&\geq \left(1 - \frac{2\Delta}{n}\right) \Pr(N \geq n - \Delta)\mathbb{E}[\mathsf{OPT}(F, n + \Delta)] \\
&\geq \left(1 - \frac{2\Delta}{n} - \Pr(N < n - \Delta)\right) \mathbb{E}[\mathsf{OPT}(F, n + \Delta)] \\
&\geq \left(1 - 2n^{-1/3} - \exp(-n^{1/3}/2)\right) \mathbb{E}[\mathsf{OPT}(F, n + \Delta)] \\
&\geq \left(1 - 4n^{-1/3}\right) \mathbb{E}[\mathsf{OPT}(F, n + \Delta)],
\end{aligned}
\tag{13}
$$

where, for the last three inequalities, we used respectively Bernoulli's inequality, Chernoff bound, then $e^{-y} \leq 1/y$.

Then, we upper bound the reward of any algorithm given the instance $(Q, m)$ as input. Let $L = \min_{i \neq j}\{|i - j| : \delta_i = 1, \delta_j = 1\}$ the smallest gap between two successive variables $Y_i$ drawn from $F$, and let $t_1 < \ldots < t_N$ the indices for which $\delta_i = 1$. We have for any algorithm $\mathsf{ALG}$ and positive integer $\ell$ that

$$\mathbb{E}[\mathsf{ALG}^{\mathcal{D}}(Q, m)] = \mathbb{E}[\mathsf{ALG}^{\mathcal{D}}(Q, m)\mathbb{1}_{N \geq n + \Delta \text{ or } L \leq \ell}] + \mathbb{E}[\mathsf{ALG}^{\mathcal{D}}(Q, m)\mathbb{1}_{N < n + \Delta, L > \ell}]. \tag{14}$$

Using Lemma A.3 and Lemma A.5, the first term can be bounded as follows

$$
\begin{aligned}
\mathbb{E}[\mathsf{ALG}^{\mathcal{D}}(Q, m)\mathbb{1}_{N \geq n + \Delta \text{ or } L \leq \ell}] &\leq \mathbb{E}[\mathsf{OPT}(F, N)\mathbb{1}_{N \geq n + \Delta \text{ or } L \leq \ell}] \\
&\leq \mathbb{E}[\mathsf{OPT}(F, N)\mathbb{1}_{N \geq n + \Delta}] + \mathbb{E}[\mathsf{OPT}(F, N)\mathbb{1}_{L \leq \ell}] \\
&\leq \left(\frac{6}{n^{1/3}} + 7m\ell\varepsilon^2\right) \mathbb{E}[\mathsf{OPT}(F, n + n^{2/3})].
\end{aligned}
$$

Recalling that $\varepsilon = m/n$ and taking $\ell = \sqrt{m}$, we obtain

$$\mathbb{E}[\mathsf{ALG}^{\mathcal{D}}(Q, m)\mathbb{1}_{N \geq n + \Delta \text{ or } L \leq \ell}] \leq \left(\frac{6}{n^{1/3}} + \frac{7n^2}{\sqrt{m}}\right) \mathbb{E}[\mathsf{OPT}(F, n + n^{2/3})]. \tag{15}$$

Regarding the second term in Equation (14), let $\tau$ be the stopping time of $\mathsf{ALG}$ and $t_\rho = \max\{j \le \tau : \delta_j = 1\}$ the last value sampled from $F$ and observed by $\mathsf{ALG}$ before it stops. We have

$$\mathbb{E}[\mathsf{ALG}^{\mathcal{D}}(Q,m)\mathbb{1}_{N<n+\Delta, L>\ell}] \le \mathbb{E}[\mathsf{ALG}^{\mathcal{D}}(Q,m) \mid N < n+\Delta, L > \ell]$$
$$= \mathbb{E}[\max_{i \in [\tau]} D_{\tau-i}(Y_i) \mid N < n+\Delta, L > \ell]$$
$$= \mathbb{E}[\max_{j \in [\rho]} D_{\tau-t_j}(Y_i) \mid N < n+\Delta, L > \ell]$$
$$\le \mathbb{E}[\max_{j \in [\rho]} D_{t_\rho-t_j}(Y_i) \mid N < n+\Delta, L > \ell]$$
$$= \mathbb{E}[\max\{Y_{t_\rho}, \max_{j<\rho} D_{t_\rho-t_j}(Y_{t_j})\} \mid N < n+\Delta, L > \ell]$$
$$\le \mathbb{E}[\max\{Y_{t_\rho}, \max_{j<\rho} D_\ell(Y_{t_j})\} \mid N < n+\Delta] .$$

We then prove that the last term is the reward of an algorithm $\mathsf{A}_m$ in the $D_\ell$-prophet inequality. Let us $\mathsf{A}_m$ be the algorithm that takes as input an instance $X_1, \ldots, X_{n+\Delta-1}$ of $n+\Delta$ IID random variables, then simulates $\mathsf{ALG}^{\mathcal{D}}(Q,m) \mid N < n+\Delta$ as follows: let $\delta_1, \ldots, \delta_m \overset{\text{iid}}{\sim} \mathcal{B}(n/m)$ set $N_\mathsf{A} = 0$ and for each $i \in [m]$

- if $\delta_i = 0$: $\mathsf{ALG}$ observes the value $Y_i = 0$,
- if $\delta_i = 1$: increment $N$, then $\mathsf{A}_m$ observes the next value $X_k$, and $\mathsf{ALG}$ observes $Y_i = X_k$,
- if $N_\mathsf{A} = n+\Delta-1$ or $\mathsf{ALG}$ stops, then $\mathsf{A}_m$ also stops.

When $\mathsf{ALG}$ decides to stop, the current value observed by $\mathsf{A}_m$ is $X_\rho$: the last value $Y_{t_\rho}$ observed by $\mathsf{ALG}$ such that $\delta_{t_\rho} = 0$. Observe that stopping when $N_\mathsf{A} = n+\Delta+1$, is equivalent to letting $\mathsf{ALG}$ observe zero values until the end, and stopping when $\mathsf{ALG}$ stops. Hence, the variables $Y_1, \ldots, Y_m$ have the same distribution as $m$ IID samples from $Q$ conditional to $N < n+\Delta$. Denoting $\rho$ the stopping time of $\mathsf{A}_m$ and $\epsilon_\ell(F, n+\Delta)$ as defined in Lemma A.1, we deduce that

$$\mathbb{E}[\mathsf{ALG}^{\mathcal{D}}(Q,m) \mid N < n+\Delta, L > \ell] \le \mathbb{E}[\max\{Y_{t_\rho}, \max_{j<\rho} D_\ell(Y_{t_j})\} \mid N < n+\Delta]$$
$$= \mathbb{E}[\max\{X_\rho, \max_{j<\rho} D_\ell(X_j)\}]$$
$$= \mathbb{E}[\mathsf{A}_m^{D_\ell}(F, n+\Delta)]$$
$$\le \mathbb{E}[\mathsf{A}_m^{D_\infty}(F, n+\Delta)] + \epsilon_\ell(F, n+\Delta) . \quad (16)$$

Substituting (15) and (16) in (14), with $\ell = \sqrt{m}$, yields

$$\mathbb{E}[\mathsf{ALG}^{\mathcal{D}}(Q,m)] \le \left(\frac{6}{n^{1/3}} + \frac{7n^2}{\sqrt{m}}\right) \mathbb{E}[\mathsf{OPT}(F, n+\Delta)] + \mathbb{E}[\mathsf{A}_m^{D_\infty}(F, n+\Delta)] + \epsilon_{\sqrt{m}}(F, n+\Delta) ,$$

and using Inequality 13, it follows that

$$\mathsf{CR}^{\mathcal{D}}(\mathsf{ALG}) \le \frac{\mathbb{E}[\mathsf{ALG}^{\mathcal{D}}(Q,m)]}{\mathbb{E}[\mathsf{OPT}(Q,m)]}$$
$$\le \frac{\frac{6}{n^{1/3}} + \frac{7n^2}{\sqrt{m}}}{1 - \frac{4}{n^{1/3}}} + \frac{\mathbb{E}[\mathsf{A}_m^{D_\infty}(F, n+\Delta)] + \epsilon_{\sqrt{m}}(F, n+\Delta)}{(1 - \frac{4}{n^{1/3}})\mathbb{E}[\mathsf{OPT}(F, n+\Delta)]}$$
$$\le \frac{\frac{6}{n^{1/3}} + \frac{7n^2}{\sqrt{m}}}{1 - \frac{4}{n^{1/3}}} + \frac{1}{1 - \frac{4}{n^{1/3}}} \left(\frac{\epsilon_{\sqrt{m}}(F, n+\Delta)}{\mathbb{E}[\mathsf{OPT}(F, n+\Delta)]} + \sup_{\mathsf{A:algo}} \frac{\mathbb{E}[\mathsf{A}^{D_\infty}(F, n+\Delta)]}{\mathbb{E}[\mathsf{OPT}(F, n+\Delta)]}\right) ,$$

taking the limit $m \to \infty$ and using Lemma A.1 gives

$$\mathsf{CR}^{\mathcal{D}}(\mathsf{ALG}) \leq \frac{\frac{6}{n^{1/3}}}{1 - \frac{4}{n^{1/3}}} + \frac{1}{1 - \frac{4}{n^{1/3}}} \left( \sup_{\mathsf{A:algo}} \frac{\mathbb{E}[\mathsf{A}^{D_\infty}(F, n + \Delta)]}{\mathbb{E}[\mathsf{OPT}(F, n + \Delta)]} \right)$$

$$= \frac{6}{n^{1/3} - 4} + \left( 1 + \frac{4}{n^{1/3} - 4} \right) \left( \sup_{\mathsf{A:algo}} \frac{\mathbb{E}[\mathsf{A}^{D_\infty}(F, n + \Delta)]}{\mathbb{E}[\mathsf{OPT}(F, n + \Delta)]} \right)$$

$$\leq \frac{10}{n^{1/3} - 4} + \sup_{\mathsf{A:algo}} \frac{\mathbb{E}[\mathsf{A}^{D_\infty}(F, n + n^{2/3})]}{\mathbb{E}[\mathsf{OPT}(F, n + n^{2/3})]} .$$

where the last inequality holds because $\mathbb{E}[\mathsf{A}^{D_\infty}(F, n + \Delta)] \leq \mathbb{E}[\mathsf{OPT}(F, n + \Delta)]$ for any algorithm $\mathsf{A}$. From here, the statement of the theorem can be deduced by observing that, for $k = n + n^{2/3}$, we have $n \geq (n + n^{2/3})/2 = k/2$, thus $n^{1/3} \geq k^{1/3}/2$, and we obtain

$$\mathsf{CR}^{\mathcal{D}}(\mathsf{ALG}) \leq \frac{20}{k^{1/3} - 8} + \sup_{\mathsf{A:algo}} \frac{\mathbb{E}[\mathsf{A}^{D_\infty}(F, k)]}{\mathbb{E}[\mathsf{OPT}(F, k)]}$$

$$= \sup_{\mathsf{A:algo}} \frac{\mathbb{E}[\mathsf{A}^{D_\infty}(F, k)]}{\mathbb{E}[\mathsf{OPT}(F, k)]} + O\left( \frac{1}{k^{1/3}} \right) .$$

$\square$

# B  From $D_\infty$-prophet to the $\gamma_{\mathcal{D}}$-prophet inequality

## B.1  Proof of Lemma 3.1

*Proof.* Let $\mathsf{ALG}$ be any rational algorithm in the $D_\infty$-prophet inequality. If $\mathsf{ALG}$ stops at some step $\tau \in [n]$, then by definition we have that $X_\tau > D_\infty(\max_{j<\tau} X_j)$, and thus $\mathsf{ALG}^{D_\infty}(X_1, \dots, X_n) = \mathsf{ALG}^0(X_1, \dots, X_n)$. Otherwise, if it stops at $\tau = n + 1$, then its reward is $\max_{i \in [n]} D_\infty(X_i) = D_\infty(\max_{i \in [n]} X_i)$, because $D_\infty$-is non increasing.

On the other hand, let $\mathsf{A}_*$ be the optimal dynamic programming algorithm for the $D_\infty$-prophet inequality. At any step $i$, if $X_i < D_\infty(\max_{j<i} X_j)$, then stopping at $i$ gives a reward of $D_\infty(\max_{j<i} X_j)$, while by rejecting $X_i$, the final reward is guaranteed to be at least $D_\infty(\max_{j<i} X_j)$. Thus rejecting $X_i$ can only increase the reward, it is therefore the optimal decision. $\square$

## B.2  Proof of Proposition 3.2

*Proof.* Let us place ourselves in any order model, or in the IID model. Assume that $\inf_{x>0} \frac{D_\infty(x)}{x} = 0$, then there exist a sequence $(s_k)_{k\geq 1}$ such that $\lim_{k\to\infty} \frac{D_\infty(s_k)}{s_k} = 0$.

Let $I = (F_1, \dots, F_n)$ an instance of non-negative random variables with finite expectation, and $X_i \sim F_i$ for all $i \in [n]$. Let $\mathsf{ALG}$ be a rational algorithm for the $D_\infty$-prophet inequality and let us denote $\tau$ its stopping time. Denoting $X^* := \max_{i \in [n]} X_i$ and using Lemma 3.1, we have for any constant $C > 0$ that that

$$\mathbb{E}[\mathsf{ALG}^{D_\infty}(I)] = \mathbb{E}[\mathsf{ALG}^0(I)\mathbb{1}_{\tau \leq n}] + \mathbb{E}[D_\infty(X^*)\mathbb{1}_{\tau=n+1}]$$

$$\leq \mathbb{E}[\mathsf{ALG}^0(I)] + \mathbb{E}[D_\infty(C)] + \mathbb{E}[D_\infty(X^*)\mathbb{1}_{X^*>C}]$$

$$\leq \sup_{\mathsf{A}} \mathbb{E}[\mathsf{A}^0(I)] + \mathbb{E}[D_\infty(C)] + \mathbb{E}[X^*\mathbb{1}_{X^*>C}] . \tag{17}$$

Let $k \geq 1$ a positive integer, $M$ a positive constant, and consider the instance $I^k$ of random variables $(X_1^k, \dots, X_n^k)$ with $X_i^k \sim \frac{s_k}{M} X_i$ for all $i \in [n]$. We have that $\mathsf{OPT}(I^k) = \frac{s_k}{M}\mathsf{OPT}(I)$ and $\sup_{\mathsf{A}} \mathbb{E}[\mathsf{A}^0(I^k)] = \frac{s_k}{M} \sup_{\mathsf{A}} \mathbb{E}[\mathsf{A}^0(I)]$. Therefore, using Inequality (17) with $I^k$ instead of $I$ and

$C = \frac{s_k}{M}$, then dividing by $\mathsf{OPT}(I^k)$ gives that

$$\mathsf{CR}^{D_\infty}(\mathsf{ALG}) \le \frac{\mathbb{E}[\mathsf{ALG}^{D_\infty}(I^k)]}{\mathbb{E}[\mathsf{OPT}(I^k)]} \le \sup_{\mathsf{A}} \frac{\mathbb{E}[\mathsf{A}^0(I)]}{\mathbb{E}[\mathsf{OPT}(I)]} + \frac{D_\infty(s_k)}{\frac{s_k}{M}\mathbb{E}[\mathsf{OPT}(I)]} + \frac{\mathbb{E}[\frac{s_k}{M}X^*\mathbb{1}_{X^*>M}]}{\frac{s_k}{M}\mathbb{E}[\mathsf{OPT}(I)]}$$

$$= \sup_{\mathsf{A}} \frac{\mathbb{E}[\mathsf{A}^0(I)]}{\mathbb{E}[\mathsf{OPT}(I)]} + \left(\frac{M}{\mathbb{E}[\mathsf{OPT}(I)]}\right)\frac{D_\infty(s_k)}{s_k} + \frac{\mathbb{E}[X^*\mathbb{1}_{X^*>M}]}{\mathbb{E}[\mathsf{OPT}(I)]} ,$$

and taking the limit $k \to \infty$, we obtain

$$\mathsf{CR}^{D_\infty}(\mathsf{ALG}) \le \sup_{\mathsf{A}} \frac{\mathbb{E}[\mathsf{A}^0(I)]}{\mathbb{E}[\mathsf{OPT}(I)]} + \frac{\mathbb{E}[X^*\mathbb{1}_{X^*>M}]}{\mathbb{E}[\mathsf{OPT}(I)]} ,$$

and since $X^*$ has finite expectation, taking the limit $M \to \infty$ gives

$$\mathsf{CR}^{D_\infty}(\mathsf{ALG}) \le \sup_{\mathsf{A}} \frac{\mathbb{E}[\mathsf{A}^0(I)]}{\mathbb{E}[\mathsf{OPT}(I)]} .$$

Finally, taking the infimum over all instances, we obtain that

$$\mathsf{CR}^{D_\infty}(\mathsf{ALG}) \le \sup_{\mathsf{A}} \mathsf{CR}^0(\mathsf{A}) .$$

As in the proof of Corollary 2.1.1, permuting $\inf_I$ and $\sup_{\mathsf{A}}$ is possible because there is an algorithm (dynamic programming) achieving the supremum for any instance. By Lemma 3.1, the inequality above holds for in particular for the optimal dynamic programming algorithm, which has a maximal competitive ratio. Therefore, the inequality remains true for any other algorithm $\mathsf{A}$, not necessarily rational. □

## B.3 Proof of Proposition 3.3

*Proof.* Let us place ourselves in any order model or in the IID model. Since $\gamma = \inf_{x>0} \frac{D_\infty(x)}{x}$, there exists a sequence $(s_k)_{k\ge 1}$ of positive numbers such that $\lim_{k\to\infty} \frac{D_\infty(s_k)}{s_k} = \gamma$.

For the random variables $X_1, \ldots, X_n$ taking values in $\{0, a, b\}$ a.s., in any order model, it is clear that the optimal decision when observing a zero value is to reject it, and when observing the value $b$ is to accept it. Let $\mathsf{ALG}$ be an algorithm satisfying this property and let $\tau$ be its stopping time. If $\tau = n + 1$ then necessarily $\max_{i\in[n]} X_i \ne b$, and the reward of $\mathsf{ALG}$ in that case is $D_\infty(a)$ if $\max_{i\in[n]} X_i$ and 0 otherwise. In particular, $\mathsf{ALG}$ is rational in the $D_\infty$-prophet inequality and we have by Lemma 3.1 that

$$\mathbb{E}[\mathsf{ALG}^{D_\infty}(I)] = \mathbb{E}[\mathsf{ALG}^0(I)] + \mathbb{E}[D_\infty(\max_{i\in[n]} X_i)\mathbb{1}_{\tau=n+1}]$$

$$= \mathbb{E}[\mathsf{ALG}^0(I)] + D_\infty(a)\Pr(\tau = n+1, \max_{i\in[n]} X_i = a) . \tag{18}$$

Consider now the instance $I^k$ of random variables $(X_1^k, \ldots, X_n^k)$ where $X_i^k = \frac{s_k}{a} X_i$ for all $i \in [n]$. $I^k$ satisfies the same assumptions as $I$ with $a^k = s_k$ and $b^k = \frac{s_k b}{a}$, and we have $\mathbb{E}[\mathsf{OPT}(I^k)] = \frac{s_k}{a}\mathbb{E}[\mathsf{OPT}(I)]$, $\mathbb{E}[\mathsf{ALG}^0(I^k)] \le \frac{s_k}{a}\sup_{\mathsf{A}}\mathbb{E}[\mathsf{A}^0(I)]$ and $(\max_{i\in[n]} X_i^k = a^k) \iff (\max_{i\in[n]} X_i = a)$. It follows that

$$\frac{\mathbb{E}[\mathsf{ALG}^{D_\infty}(I^k)]}{\mathbb{E}[\mathsf{OPT}(I^k)]} \le \frac{\sup_{\mathsf{A}}\mathbb{E}[\mathsf{A}^0(I)]}{\mathbb{E}[\mathsf{OPT}(I)]} + \frac{D_\infty(s_k)}{\frac{s_k}{a}\mathbb{E}[\mathsf{OPT}(I)]}\Pr(\tau = n+1, \max_{i\in[n]} X_i = a)$$

$$= \frac{\sup_{\mathsf{A}}\mathbb{E}[\mathsf{A}^0(I)]}{\mathbb{E}[\mathsf{OPT}(I)]} + \left(\frac{D_\infty(s_k)}{s_k}\right)\frac{a}{\mathbb{E}[\mathsf{OPT}(I)]}\Pr(\tau = n+1, \max_{i\in[n]} X_i = a) .$$

Taking the limit $k \to \infty$ gives

$$\mathsf{CR}^{D_\infty}(\mathsf{ALG}) \le \frac{\sup_{\mathsf{A}}\mathbb{E}[\mathsf{A}^0(I)] + \gamma a \Pr(\tau = n+1, \max_{i\in[n]} X_i = a)}{\mathbb{E}[\mathsf{OPT}(I)]}$$

$$= \frac{\sup_{\mathsf{A}}(\mathbb{E}[\mathsf{A}^0(I)] + \mathbb{E}[\gamma(\max_{i\in[n]} X_i)\mathbb{1}_{\tau=n+1}])}{\mathbb{E}[\mathsf{OPT}(I)]} .$$

ALG is also rational in the $\gamma$-prophet inequality. Therefore, using Lemma 3.1, we deduce that

$$\mathsf{CR}^{D\infty}(\mathsf{ALG}) \leq \frac{\mathbb{E}[\mathsf{ALG}^\gamma(I)]}{\mathbb{E}[\mathsf{OPT}(I)]} \leq \sup_{\mathsf{A}} \frac{\mathbb{E}[\mathsf{A}^\gamma(I)]}{\mathbb{E}[\mathsf{OPT}(I)]}.$$

This upper bound is true for the optimal dynamic programming algorithm, since it rejects all zeros and accepts $b$, therefore the upper bound also holds for any other algorithm. $\qquad\square$

## C   The $\gamma_{\mathcal{D}}$-prophet inequality

### C.1   Proof of Proposition 4.2

*Proof.* For the lower bound, it suffices to consider the following trivial algorithm: if $\alpha > \gamma$ then run $\mathsf{A}_\alpha$, and if $\gamma > \alpha$ then observe all the items then select the one with maximum value.

For the upper bound, let $I = (F_1, \ldots, F_n)$ be an instance of the problem and $X_i \sim F_i$ for all $i$, and let $\beta_I := \sup_{\mathsf{A}} \frac{\mathbb{E}[\mathsf{A}^0(I)]}{\mathbb{E}[\mathsf{OPT}(I)]}$. Let $\mathsf{A}$ be any algorithm, $\tau$ its stopping time, and $Y_\tau = \max_{i<\tau} X_{\pi(i)}$, where $\pi$ is the observation order. With the previous notations, we can write that $\mathbb{E}[\mathsf{A}^\gamma(I)] = \mathbb{E}[\max(X_{\pi(\tau)}, \gamma Y_\tau)]$. For any $x, y \geq 0$, the application $\gamma \mapsto \max(x, \gamma y)$ is convex on $[0, 1]$, hence it can be upper bounded by $(1 - \gamma)x + \gamma \max(x, y)$. Therefore

$$\mathbb{E}[\mathsf{A}^\gamma(I)] \leq (1 - \gamma)\mathbb{E}[X_{\pi(\tau)}] + \gamma\mathbb{E}[\max(X_{\pi(\tau)}, Y_\tau)]$$
$$\leq (1 - \gamma)\mathbb{E}[\mathsf{A}^0(I)] + \gamma\mathbb{E}[\mathsf{OPT}(I)]$$
$$\leq \big((1 - \gamma)\beta_I + \gamma\big)\mathbb{E}[\mathsf{OPT}(I)] .$$

Therefore, $\mathsf{CR}^\gamma(\mathsf{ALG}) \leq ((1 - \gamma)\beta_I + \gamma)$. Taking the infimum over all the instances $I$ gives the result. Indeed, if we denote $\mathsf{A}_*$ the optimal dynamic programming algorithm for the standard prophet inequality, then

$$\inf_I \beta_I = \inf_I \frac{\mathbb{E}[\mathsf{A}_*^0(I)]}{\mathbb{E}[\mathsf{OPT}(I)]} = \mathsf{CR}^0(\mathsf{A}_*) \leq \beta .$$

$\qquad\square$

### C.2   Proofs for the adversarial order model

*Proof.* We first prove the upper bound, and then analyze the single threshold algorithm proposed in the theorem.

**Upper bound**   Let $\varepsilon \in (0, 1 - \gamma)$, and let $a = \frac{1}{1-(1-\varepsilon)\gamma}$ (such that $1 + (1 - \varepsilon)\gamma a = a$). Let $X_1, X_2$ the two random variables defined by $X_1 = a$ almost surely and

$$X_2 = \begin{cases} \frac{1}{\varepsilon} & \text{w.p. } \varepsilon \\ 0 & \text{w.p. } 1 - \varepsilon \end{cases} .$$

Stopping at the first step gives a reward of $a$, while stopping at the second step gives

$$\mathbb{E}[\max(\gamma a, X_2)] = \varepsilon \times \frac{1}{\varepsilon} + (1 - \varepsilon) \times \gamma a = 1 + (1 - \varepsilon)\gamma a = a ,$$

hence the expected output of any algorithm is exactly equal to $a$. On the other hand

$$\mathbb{E}[\max(X_1, X_2)] = 1 + (1 - \varepsilon)a ,$$

and we deduce that, for any algorithm $\mathsf{ALG}$ for the $\gamma$-prophet inequality, we have

$$\mathsf{CR}(\mathsf{ALG}) \leq \frac{\mathbb{E}[\mathsf{ALG}(X_1, X_2)]}{\mathbb{E}[\max(X_1, X_2)]} = \frac{a}{1 + (1 - \varepsilon)a} ,$$

and this is true for any $\varepsilon \in (0, 1 - \gamma)$, thus taking $\varepsilon \to 0$ gives

$$\mathsf{CR}(\mathsf{ALG}) \leq \frac{\frac{1}{1-\gamma}}{1 + \frac{1}{1-\gamma}} = \frac{1}{2 - \gamma} .$$

**Lower bound** Consider an algorithm with an acceptance threshold $\theta$, i.e. that accepts the first value larger than $\theta$. Let $I = (F_1, \ldots, F_n)$ be any instance, such that $X_i \sim F_i$ for all $i$, and let us define $X^* = \max_{i \in [n]} X_i$ and $p = \Pr(X^* \leq \theta)$. In the classical prophet inequality, if no value is larger than $\theta$ then the reward of the algorithm is zero, and we have the classical lower bound

$$\mathbb{E}[\mathsf{ALG}^0(I)] \geq (1-p)\theta + p\mathbb{E}[(X^* - \theta)_+],$$

For the $\gamma$-prophet, if no value is larger than $\theta$ (i.e $X^* \leq \theta$), then the algorithm gains $\gamma X^*$ instead of 0. Therefore, it holds that

$$
\begin{aligned}
\mathbb{E}[\mathsf{ALG}^\gamma(I)] &= \mathbb{E}[\mathsf{ALG}^0(I)] + \mathbb{E}[X^* \mathbb{1}_{X^* \leq \theta}] \\
&\geq (1-p)\theta + p\mathbb{E}[(X^* - \theta)_+] + \gamma\mathbb{E}[X^* \mathbb{1}_{X^* \leq \theta}] \\
&= (1-p)\theta + p\mathbb{E}[(X^* - \theta)\mathbb{1}_{X^* > \theta}] + \gamma\mathbb{E}[X^* \mathbb{1}_{X^* \leq \theta}] \\
&= (1-p)\theta + p(\mathbb{E}[X^* \mathbb{1}_{X^* > \theta}] - (1-p)\theta) + \gamma\mathbb{E}[X^* \mathbb{1}_{X^* \leq \theta}] \\
&= (1-p)^2\theta + p\mathbb{E}[X^* \mathbb{1}_{X^* > \theta}] + \gamma\mathbb{E}[X^* \mathbb{1}_{X^* \leq \theta}] ,
\end{aligned}
$$

and observing that

$$\theta = \frac{\mathbb{E}[\theta \mathbb{1}_{X^* \leq \theta}]}{p} \geq \frac{\mathbb{E}[X^* \mathbb{1}_{X^* \leq \theta}]}{p} ,$$

we deduce the lower bound

$$\mathbb{E}[\mathsf{ALG}^\gamma(I)] \geq p\mathbb{E}[X^* \mathbb{1}_{X^* > \theta}] + \left(\gamma + \frac{(1-p)^2}{p}\right)\mathbb{E}[X^* \mathbb{1}_{X^* \leq \theta}]$$

$$\geq \min\left\{p, \gamma + \frac{(1-p)^2}{p}\right\}\mathbb{E}[X^*] .$$

The right term is maximized for $p$ satisfying $p = \gamma + \frac{(1-p)^2}{p}$, that leads to

$$p = \gamma + \frac{(1-p)^2}{p} \iff p^2 = \gamma p + 1 - 2p + p^2$$

$$\iff p = \frac{1}{2 - \gamma} .$$

Hence, by choosing a threshold $\theta$ satisfying $\Pr(X^* \leq \theta) = \frac{1}{2-\gamma}$ we obtain a competitive ratio of at least $\frac{1}{2-\gamma}$. $\qquad\square$

### C.3 Proofs for the random order model

We prove here the upper and lower bounds stated in Theorem 4.4.

#### C.3.1 Proof of Theorem 4.4

*Proof.* We first prove the upper bound, and then derive the analysis for single threshold algorithms.

**Upper bound** Let $a > 0$, and let $X_1, \ldots, X_{n+1}$ be independent random variables such that $X_{n+1} = a$ a.s. and for $1 \leq i \leq n$

$$X_i \sim \begin{cases} n & \text{w.p. } \frac{1}{n^2} \\ 0 & \text{w.p. } 1 - \frac{1}{n^2} \end{cases} .$$

Any reasonable algorithm skips zero values and stops when observing the value $n$. The only strategic decision to make is thus to stop or not when observing $X_{n+1} = a$. By analyzing the dynamic programming solution $\mathsf{ALG}_\star$ we obtain that the optimal decision rule is to skip $a$ if it is observed before a certain step $j$, and select it otherwise. The step $j$ corresponds to the time when the expectation of the future reward is less than $a$. Let $\pi$ be the random order in which the variables are observed. Then, if $\pi^{-1}(n + 1) < j$, i.e. if the value $a$ is observed before time $j$, $X_{n=1}$ is not selected. The output of this algorithm is hence $n$ if at least one random variable equals $n$, and $\gamma a$ otherwise. This leads to

$$\mathbb{E}[\mathsf{ALG}_\star^\gamma(X) \mid \pi^{-1}(n + 1) < j] = n\left(1 - \left(1 - \frac{1}{n^2}\right)^n\right) + \gamma a \left(1 - \frac{1}{n^2}\right)^n$$

$$\leq 1 + \gamma a ,$$

where we used the inequality $\left(1 - \frac{1}{n^2}\right)^n \geq 1 - \frac{1}{n}$. On the other hand, if $\pi^{-1}(n) \geq j$, then $a$ is selected if the value $n$ has not been observed before it, hence for any $i \geq j$ we have

$$\mathbb{E}[\mathsf{ALG}_\star^\gamma(X) \mid \pi^{-1}(n+1) = i] = n\left(1 - \left(1 - \frac{1}{n^2}\right)^{i-1}\right) + a\left(1 - \frac{1}{n^2}\right)^{i-1}$$

$$\leq \frac{i-1}{n} + a ,$$

we deduce that

$$\mathbb{E}[\mathsf{ALG}_\star^\gamma(X)] \leq (1 + \gamma a)\Pr(\pi^{-1}(n) \leq j - 1) + \sum_{i=j}^{n+1}\left(\frac{i-1}{n} + a\right)\Pr(\pi^{-1}(n) = i)$$

$$= \frac{j-1}{n+1}(1 + \gamma a) + \frac{1}{n+1}\sum_{i=j}^{n+1}\left(\frac{i-1}{n} + a\right)$$

$$= (1 - (1-\gamma)a)\frac{j}{n} - \frac{1}{2}\left(\frac{j}{n}\right)^2 + \frac{1}{2} + a + o(1)$$

$$\leq 1 + 2\gamma a + (1 - \gamma)^2 a^2 + o(1) ,$$

where the last inequality is obtained by maximizing over $j/n$. Finally, we directly obtain that

$$\mathbb{E}[\max_i X_i] = n\left(1 - \left(1 - \frac{1}{n^2}\right)^n\right) + a\left(1 - \frac{1}{n^2}\right)^n$$

$$= 1 + a + o(1) ,$$

so for any algorithm $\mathsf{ALG}$ we obtain that

$$\mathsf{CR}^\gamma(\mathsf{ALG}) \leq \mathsf{CR}^\gamma(\mathsf{ALG}_\star) \leq \frac{1 + 2\gamma a + (1-\gamma)^2 a^2}{1 + a} .$$

The function above is minimized for $a = \sqrt{\frac{3-\gamma}{1-\gamma}} - 1$, which translates to

$$\mathsf{CR}^\gamma(\mathsf{ALG}) \leq (1-\gamma)^{3/2}(\sqrt{3-\gamma} - \sqrt{1-\gamma}) + \gamma .$$

**Lower bound** We still denote by $I = (F_1, \dots, F_n)$ the input instance and $X_i \sim F_i$ for all $i \in [n]$. Let $\mathsf{ALG}$ be the algorithm with single threshold $\theta$, then it is direct that

$$\mathsf{ALG}^\gamma(I) = \mathsf{ALG}^0(I) + \gamma X^* \mathbb{1}_{X^* < \theta} . \tag{19}$$

We start by giving a lower bound on $\mathbb{E}[\mathsf{ALG}^0]$. We use from Correa et al. [2021c] (Theorem 2.1) that for any $x < \theta$ it holds that

$$\Pr(\mathsf{ALG}^0(I) \geq x) = \Pr(\mathsf{ALG}^0(I) \geq \theta) = \Pr(X^* \geq \theta) = 1 - p ,$$

and for $x \geq \theta$ it holds that

$$\Pr(\mathsf{ALG}^0(I) \geq x) \geq \frac{1-p}{-\log p}\Pr(X^* \geq x) ,$$

from which we deduce that

$$\mathbb{E}[\mathsf{ALG}^0(I)] = \int_0^\infty \Pr(\mathsf{ALG}^0(I) \geq x)dx$$

$$\geq (1-p)\theta + \frac{1-p}{-\log p}\int_\theta^\infty \Pr(X^* \geq x)dx . \tag{20}$$

On the other hand, we obtain that

$$\mathbb{E}[X^* \mathbb{1}_{X^* < \theta}] = \int_0^\infty \Pr(X^* \mathbb{1}_{X^* < \theta} \geq x)dx = \int_0^\infty \Pr(x \leq X^* < \theta)dx$$

$$= \int_0^\theta (\Pr(X^* > x) - \Pr(X^* \geq \theta))dx$$

$$= \int_0^\theta \Pr(X^* > x)dx - (1-p)\theta . \tag{21}$$

Using (19), (20) and (21) we deduce that

$$
\begin{aligned}
\mathbb{E}[\mathsf{ALG}^\gamma(I)] &\geq (1-p)\theta + \frac{1-p}{-\log p} \int_\theta^\infty \Pr(X^* \geq x)dx + \gamma \int_0^\theta \Pr(X^* \geq x)dx - \gamma(1-p)\theta \\
&= (1-\gamma)(1-p)\theta + \gamma \int_0^\theta \Pr(X^* \geq x)dx + \frac{1-p}{-\log p} \int_\theta^\infty \Pr(X^* \geq x)dx \\
&\geq ((1-\gamma)(1-p)+\gamma) \int_0^\theta \Pr(X^* \geq x)dx + \frac{1-p}{-\log p} \int_\theta^\infty \Pr(X^* \geq x)dx \\
&\geq \min\left\{ (1-\gamma)(1-p)+\gamma, \, \frac{1-p}{-\log p} \right\} \left( \int_0^\theta \Pr(X^* \geq x)dx + \int_\theta^\infty \Pr(X^* \geq x)dx \right) \\
&= \min\left\{ 1-(1-\gamma)p, \, \frac{1-p}{-\log p} \right\} \mathbb{E}[X^*] \,.
\end{aligned}
$$

Finally, choosing $p = p_\gamma$ gives the result. $\qquad\square$

### C.3.2 Proof of Corollary 4.4.1

*Proof.* For $p = \frac{1/e}{1-(1-1/e)\gamma}$, we have immediately that

$$
1-(1-\gamma)p = 1 - \frac{(1-\gamma)/e}{1-(1-1/e)\gamma} \,,
$$

and $p \in [1/e, 1]$ for any $\gamma \in [0,1]$. Since the function $x \mapsto (1-x)/\log(1/x)$ is concave, we can lower bound it on $[1/e, 1]$ by $x \mapsto 1 - 1/e + \frac{x-1/e}{e-1}$, which is the line intersecting it in $1/e$ and $1$. Therefore we have

$$
\frac{1-p}{-\log p} \geq 1 - 1/e + \frac{p-1/e}{e-1} = 1 - \frac{(1-\gamma)/e}{1-(1-1/e)\gamma} \,.
$$

Finally, using the previous theorem, this choice of $p$ guarantees a competitive ratio of at least

$$
\min\left\{ 1-(1-\gamma)p, \, \frac{1-p}{-\log p} \right\} = 1 - \frac{(1-\gamma)/e}{1-(1-1/e)\gamma} \,.
$$

$\qquad\square$

## C.4 Proofs for the IID model

### Proof of Lemma 4.5

*Proof of Lemma 4.5.* Let $F$ be the cumulative distribution function of $X_1$, $a > 0$ and $\mathsf{ALG}$ the algorithm with single threshold $\theta$ such that $1 - F(\theta) = \frac{a}{n}$. We denote $X^* = \max_{i \in [n]} X_i$. As in the previous proofs, we will begin by lower bounding $\mathsf{ALG}^0(F, n)$. For any $i \in [n]$, $\mathsf{ALG}$ stops at step $i$ if and only if $X_i > \theta$ and all the previous items were rejected, i.e. $X_j \leq \theta$ for all $j < i$. Thus we can write

$$
\begin{aligned}
\mathbb{E}[\mathsf{ALG}^0(F,n)] &= \mathbb{E}\left[ \sum_{i=1}^n (X_i \mathbb{1}_{X_i > \theta}) \mathbb{1}_{\forall j < i : X_j \leq \theta} \right] \\
&= \sum_{i=1}^n F(\theta)^{i-1} \mathbb{E}[X_i \mathbb{1}_{X_i > \theta}] \\
&= \frac{1 - F(\theta)^n}{1 - F(\theta)} \mathbb{E}[X_1 \mathbb{1}_{X_1 > \theta}] \\
&= \frac{1 - F(\theta)^n}{a} \times n \mathbb{E}[X_1 \mathbb{1}_{X_1 > \theta}] \,,
\end{aligned}
\tag{22}
$$

where the second equality is true by the independence of the random variables $(X_i)_i$. On the other hand, we can upper bound $\mathbb{E}[X^* \mathbb{1}_{X^* > \theta}]$ as follows

$$\mathbb{E}[X^* \mathbb{1}_{X^* > \theta}] \leq \Pr(X^* > \theta)\theta + \mathbb{E}[(X^* - \theta)_+]$$

$$\leq \Pr(X^* > \theta)\theta + \mathbb{E}\Big[ \sum_{i=1}^{n}(X_i - \theta)_+ \Big]$$

$$= \Pr(X^* > \theta)\theta + n\big(\mathbb{E}[X_1 \mathbb{1}_{X_1 > \theta}] - \Pr(X_1 > \theta)\theta\big)$$

$$= (1 - F(\theta)^n - a)\theta + n\mathbb{E}[X_1 \mathbb{1}_{X_1 > \theta}] .$$

Using the definition of $\theta$, the independence of $(X_i)_i$ then Bernoulli's inequality we have that

$$\Pr(X^* > \theta) = 1 - F(\theta)^n = 1 - \Big(1 - \frac{a}{n}\Big)^n \leq 1 - (1 - n \times \frac{a}{n}) = a ,$$

and observing that $\theta = \frac{\mathbb{E}[\theta \mathbb{1}_{X^* \leq \theta}]}{F(\theta)^n} \geq \frac{\mathbb{E}[X^* \mathbb{1}_{X^* \leq \theta}]}{F(\theta)^n}$, we deduce that

$$\mathbb{E}[X^* \mathbb{1}_{X^* > \theta}] \leq - \Big(1 - \frac{1-a}{F(\theta)^n}\Big) \mathbb{E}[X^* \mathbb{1}_{X^* \leq \theta}] + n\mathbb{E}[X_1 \mathbb{1}_{X_1 > \theta}] .$$

by substituting into (22), we obtain

$$\mathbb{E}[\mathsf{ALG}^0(F, n)] \geq \frac{1 - F(\theta)^n}{a} \left( \mathbb{E}[X^* \mathbb{1}_{X^* > \theta}] + \Big(1 - \frac{1-a}{F(\theta)^n}\Big) \mathbb{E}[X^* \mathbb{1}_{X^* \leq \theta}] \right) .$$

Finally, the reward in the $\gamma$-prophet inequality is

$$\mathbb{E}[\mathsf{ALG}^\gamma(F, n)] = \mathbb{E}[\mathsf{ALG}^0(F, n)] + \gamma\mathbb{E}[X^* \mathbb{1}_{X^* < \theta}]$$

$$\geq \frac{1 - F(\theta)^n}{a} \mathbb{E}[X^* \mathbb{1}_{X^* > \theta}] + \Big( \frac{1 - F(\theta)^n}{a} \Big(1 - \frac{1-a}{F(\theta)^n}\Big) + \gamma \Big) \mathbb{E}[X^* \mathbb{1}_{X^* < \theta}]$$

$$\geq \min \left\{ \frac{1 - F(\theta)^n}{a}, \frac{1 - F(\theta)^n}{a} \Big(1 - \frac{1-a}{F(\theta)^n}\Big) + \gamma \right\} \mathbb{E}[X^*] .$$

The equation $\frac{1-F(\theta)^n}{a} = \frac{1-F(\theta)^n}{a} \Big(1 - \frac{1-a}{F(\theta)^n}\Big) + \gamma$, is equivalent to

$$\left( \frac{1}{(1 - a/n)^n} - 1 \right) \left( \frac{1}{a} - 1 \right) = \gamma , \tag{23}$$

and for any $n \geq 2$ the function $a \mapsto \big( \frac{1}{(1-a/n)^n} - 1 \big) \big( \frac{1}{a} - 1 \big)$ is decreasing on $(0, 1]$ and goes from 1 to 0, thus Equation (23) admits a unique solution $a_{n,\gamma}$, and taking $a = a_{n,\gamma}$ guarantees a reward of $\frac{1-F(\theta)^n}{a_{n,\gamma}} \mathbb{E}[X^*] = \frac{1-(1-\frac{a_{n,\gamma}}{n})^n}{a_{n,\gamma}} \mathbb{E}[X^*]$. $\qquad \square$

**Proof of Theorem 4.6**

*Proof.* We first prove the upper bound, and then we give the single-threshold algorithm satisfying the lower bound.

**Upper bound**    We consider an instance similar to the one used in the proof of Theorem 4.4. Let $a, x > 0$, and let $X_1, \ldots, X_n$ be IID random variables with the following the distribution $F$ defined by

$$X_1 \sim \begin{cases} n & \text{w.p. } \frac{1}{n^2} \\ a & \text{w.p. } \frac{x}{n} \\ 0 & \text{w.p. } 1 - \frac{x}{n} - \frac{1}{n^2} \end{cases} .$$

A reasonable algorithm would always reject the value $0$ and accept the value $n$. However, if the algorithm faces an item with value $a$, it must decide to either accept it, or reject it with a guarantee of recovering $\gamma a$ at the end. By analyzing the dynamic programming algorithm $\mathsf{ALG}_\star$, we find that the optimal decision is to reject $a$ if observed before a certain step $j$, and accept it otherwise. Let us denote $\tau$ the stopping time of $\mathsf{ALG}_\star$. By convention, we write $\tau = n + 1$ to say that no value was selected by the algorithm, in which case the reward is $\gamma \max_{i \in [n]} X_i$.

If $\tau \leq j - 1$ then necessarily $X_\tau = n$, because $\mathsf{ALG}_\star$ rejects the value $a$ if it is met before step $j$, and if $\tau = n + 1$ then $\max_{i \in [n]} X_i \in \{0, a\}$, because otherwise the algorithm would have selected the value $n$ and stopped earlier. It follows that the expected output of $\mathsf{ALG}_\star$ on this instance is

$$\mathbb{E}[\mathsf{ALG}_\star^\gamma(F, n)] = n \Pr(\tau < j) + \sum_{i=j}^n \mathbb{E}[X_i \mid \tau = i] \Pr(\tau = i)$$
$$+ \gamma a \Pr(\tau = n + 1, \max_{i \in [n]} X_i = a) . \tag{24}$$

Let us now compute the terms above one by one.

$$\Pr(\tau < j) = \Pr(\exists i \in [j - 1] : X_i = n) = 1 - \left(1 - \frac{1}{n^2}\right)^{j-1} \leq \frac{j}{n^2} ,$$

where we used Bernoulli's inequality $(1 - 1/n^2)^{j-1} \geq 1 - \frac{j-1}{n^2} > 1 - \frac{j}{n^2}$. For $i \in \{j, \dots, n\}$, $\mathsf{ALG}_\star$ stops at $i$ if $X_i \in \{a, n\}$ and if it has not stopped before, i.e. $X_k \in \{0, a\}$ for all $k < j$ and $X_k = 0$ for all $k \in \{j, \dots, i - 1\}$, hence

$$\Pr(\tau = i) = \Pr(\forall k < j : X_k \neq n \text{ and } \forall j \leq k \leq i - 1 : X_k = 0 \text{ and } X_i \neq 0)$$
$$= \left(1 - \frac{1}{n^2}\right)^{j-1} \left(1 - \frac{x}{n} - \frac{1}{n^2}\right)^{i-j} \Pr(X_i \neq 0)$$
$$\leq \left(1 - \frac{x}{n}\right)^{i-j} \Pr(X_i \neq 0) ,$$

the second equality is true by independence, and the last inequality holds because $1 - \frac{1}{n^2} \leq 1$ and $1 - \frac{x}{n} - \frac{1}{n^2} \leq 1 - \frac{x}{n}$. By independence of the variables $(X_k)_k$, we also have that

$$\mathbb{E}[X_i \mid \tau = i] = \mathbb{E}[X_i \mid X_i \neq 0] = \frac{\mathbb{E}[X_i]}{\Pr(X_i \neq 0)} = \frac{1 + ax}{n \Pr(X_i \neq 0)} .$$

Finally, the event $(\tau = n + 1, \max_{i \in [n]} X_i = a)$ is equivalent $(\max_{i \in [j-1]} X_i = a, \forall k \geq j : X_k = 0)$. In fact, the algorithm does not stop before $n + 1$ if and only if $X_k \neq n$ for all $k < j$ and $X_k = 0$ for all $j \leq k \leq n$, and under these conditions, it holds that $\max_{i \in [n]} X_i = \max_{i \in [j-1]} X_i$. Therefore

$$\Pr(\tau = n + 1, \max_{i \in [n]} X_i = a) = \Pr(\max_{i \in [j-1]} X_i = a, \forall k \geq j : X_k = 0)$$
$$\leq \Pr(\max_{i \in [j-1]} X_i \neq 0) \Pr(\forall k \geq j : X_k = 0)$$
$$= \left(1 - \left(1 - \frac{x}{n} - \frac{1}{n^2}\right)^{j-1}\right) \left(1 - \frac{x}{n} - \frac{1}{n^2}\right)^{n-j}$$
$$= \left(1 - e^{-\frac{xj}{n}} + o(1)\right)\left(e^{-x + \frac{xj}{n}} + o(1)\right)$$
$$= \left(e^{\frac{xj}{n}} - 1\right)e^{-x} + o(1) .$$

All in all, we obtain by substituting into 24 that

$$\mathbb{E}[\mathsf{ALG}_\star^\gamma(F, n)] \leq \frac{j}{n} + \left(\frac{1 + ax}{n}\right) \sum_{i=j}^n \left(1 - \frac{x}{n}\right)^{i-j} + \gamma a e^{-x}\left(e^{\frac{xj}{n}} - 1\right) + o(1)$$
$$= \frac{j}{n} + \left(\frac{1 + ax}{n}\right) \frac{1 - (1 - x/n)^{n-j+1}}{x/n} + \gamma a e^{-x}\left(e^{\frac{xj}{n}} - 1\right) + o(1)$$
$$= \frac{j}{n} + \left(\frac{1}{x} + a\right) \left(1 - e^{-x + \frac{xj}{n}} + o(1)\right) + \gamma a e^{-x}\left(e^{\frac{xj}{n}} - 1\right) + o(1)$$
$$= \frac{j}{n} - \left[\left(\tfrac{1}{x} + (1 - \gamma)a\right)\right]e^{-x + \frac{xj}{n}} + \frac{1}{x} + a - \gamma a e^{-x} + o(1)$$
$$\leq \max_{s > 0} \left\{s - \left[\left(\tfrac{1}{x} + (1 - \gamma)a\right)\right]e^{-x + xs}\right\} + \frac{1}{x} + (1 - \gamma e^{-x})a + o(1)$$
$$= -\frac{1}{x}\left(\log(1 + (1 - \gamma)ax) + 1 - x\right) + \frac{1}{x} + (1 - \gamma e^{-x})a + o(1)$$
$$= -\frac{1}{x} \log(1 + (1 - \gamma)ax) + 1 + (1 - \gamma e^{-x})a + o(1) .$$

On the other hand, we have that

$$\Pr(\max_{i\in[n]} X_i = n) = 1 - \left(1 - \frac{1}{n^2}\right)^n = \frac{1}{n} + o(1/n) \, ,$$

$$\Pr(\max_{i\in[n]} X_i = 0) = \left(1 - \frac{x}{n} - \frac{1}{n^2}\right) = e^{-x} + o(1) \, ,$$

$$\Pr(\max_{i\in[n]} X_i = a) = 1 - \Pr(\max_{i\in[n]} X_i = 0) - \Pr(\max_{i\in[n]} X_i = n) = 1 - e^{-x} + o(1) \, ,$$

therefore, the expected maximum value is

$$\mathbb{E}[\max_{i\in[n]} X_i] = n\Pr(\max_{i\in[n]} X_i = n) + a\Pr(\max_{i\in[n]} X_i = a)$$

$$= 1 + \left(1 - e^{-x}\right)a + o(1) \, .$$

We deduce that

$$\frac{\mathbb{E}[\mathsf{ALG}^{\gamma}_{\star}(F,n)]}{\mathbb{E}[\max_{i\in[n]} X_i]} \leq \frac{-\frac{1}{x}\log(1 + (1-\gamma)ax) + 1 + (1 - \gamma e^{-x})a}{1 + \left(1 - e^{-x}\right)a} + o(1)$$

$$= 1 - \frac{\frac{1}{x}\log(1 + (1-\gamma)ax) - (1-\gamma)ae^{-x}}{1 + \left(1 - e^{-x}\right)a} + o(1) \, .$$

Consequently, for any $a, x > 0$ and for any algorithm $\mathsf{ALG}$ we have

$$\mathsf{CR}(\mathsf{ALG}) \leq \mathsf{CR}(\mathsf{ALG}_{\star})$$

$$\leq \lim_{n\to\infty} \frac{\mathbb{E}[\mathsf{ALG}^{\gamma}_{\star}(F,n)]}{\mathbb{E}[\max_{i\in[n]} X_i]}$$

$$\leq 1 - \frac{\log(1 + (1-\gamma)ax) - (1-\gamma)axe^{-x}}{x + \left(1 - e^{-x}\right)ax} \, .$$

In particular, for $x = 2$ and $a = \frac{1-\gamma/2}{1-\gamma}$ we find that

$$\mathsf{CR}(\mathsf{ALG}) \leq 1 - \frac{\log(3 - \gamma) - (2 - \gamma)e^{-2}}{2 + \frac{2-\gamma}{1-\gamma}(1 - e^{-2})}$$

$$= 1 - (1 - \gamma)\frac{e^2\log(3 - \gamma) - (2 - \gamma)}{2(2e^2 - 1) - \gamma(3e^2 - 1)}$$

$$= U(\gamma) \, .$$

This proves the upper bound stated in the theorem, and we can verify that it is increasing, and satisfies $U(0) = \frac{4-\log 3}{4-2/e^2}$ $U(1) = 1$

**Lower bound on the competitive ratio**    We will prove that the algorithm presented in Lemma 4.5 has a competitive ratio of at least $(1 - (1-\gamma)p_{\gamma})$, where $p_{\gamma}$, first introduced in Theorem 4.4, is the unique solution of the equation $(1-(1-\gamma)p) = \frac{1-p}{-\log p}$, which is equivalent to $\left(\frac{1}{p}-1\right)\left(\frac{1}{\log(1/p)}-1\right) = \gamma$.

Let $a_{\gamma} = -\log(p_{\gamma})$. It follows from the definition of $p_{\gamma}$ that $a_{\gamma}$ is the unique solution of the equation $(e^a - 1)(\frac{1}{a} - 1) = \gamma$. For any $n \geq 2$ and $x \geq 0$ we have that $(1 - x/n)^n \leq e^{-x}$, hence, by definition of $a_{n,\gamma}$ and $a_{\gamma}$

$$\left(\frac{1}{e^{-a_{n,\gamma}}} - 1\right)\left(\frac{1}{a_{n,\gamma}} - 1\right) \leq \left(\frac{1}{(1 - \frac{a_{n,\gamma}}{n})^n} - 1\right)\left(\frac{1}{a_{n,\gamma}} - 1\right)$$

$$= \gamma$$

$$= \left(\frac{1}{e^{-a_{\gamma}}} - 1\right)\left(\frac{1}{a_{\gamma}} - 1\right) \, . \tag{25}$$

Moreover, the function $x \mapsto (e^x - 1)(1/x - 1)$ is decreasing on $(0, 1)$. In fact its derivative at any point $x \in (0, 1)$ is

$$\frac{d}{dx}\left[\left(\frac{1}{e^{-x}} - 1\right)\left(\frac{1}{x} - 1\right)\right] = \left(\frac{1}{x} - 1\right)e^x - \frac{e^x - 1}{x^2}$$

$$= \frac{1}{x^2}\left(1 - x^2 - (1-x)e^x\right)$$

$$= \frac{1-x}{x^2}(1 + x - e^x) < 0 .$$

It follows from (25) that $a_\gamma \leq a_{n,\gamma}$. Finally, given that $x \mapsto \frac{1-e^{-x}}{x}$ is non-increasing on $(0, 1]$, we deduce that

$$\frac{1 - (1 - \frac{a_{n,\gamma}}{n})^n}{a_{n,\gamma}} \geq \frac{1 - e^{-a_{n,\gamma}}}{a_{n,\gamma}} \geq \frac{1 - e^{-a_\gamma}}{a_\gamma} .$$

We deduce that the competitive ratio of the algorithm described in Theorem 4.6 is at least $\frac{1-e^{-a_\gamma}}{a_\gamma} = \frac{1-p_\gamma}{\log(1/p_\gamma)} = 1 - (1 - \gamma)p_\gamma$.

$\square$

# D   Random decay functions

While we only studied deterministic decay functions in the paper, it is also possible to have scenarios with random decay functions. Consider for example that rejected items remain available after $j$ steps with a probability $p_j$, this is modeled by $D_j(x) = \xi_j x$ with $\xi_j$ a Bernoulli random variable with parameter $p_j$. We explain in this section how the definitions and our results extend to this case.

**Definition D.1** (Random process). *Let $\mathcal{X}$ is a non-empty set. A random process o $\mathcal{X}$ is a collection of random variables $\{Z_x\}_{x \in \mathcal{X}}$. Two random processes $\mathcal{Z} = \{Z_x\}_{x \in \mathcal{X}}$ and $\mathcal{Z}' = \{Z'_x\}_{x \in \mathcal{X}'}$ are independent if any finite sub-process of $\mathcal{Z}$ is independent of any sub-process of $\mathcal{Z}'$. For simplicity, let us say that the random processes $\{Z_x^1\}_{x \in \mathcal{X}^1}, \dots, \{Z_x^m\}_{x \in \mathcal{X}^m}$ are mutually independent if, for any $x_1 \in \mathcal{X}_1, \dots, x_m \in \mathcal{X}_m$, the random variables $Z_{x_1}^1, \dots, Z_{x_m}^m$ are mutually independent.*

**Definition D.2** (Random decay functions). *Let $\mathcal{D} = (D_1, D_2, \dots)$ be a sequence of mutually independent random processes. We say that $\mathcal{D}$ is a sequence of random decay functions if*

1. *$\Pr(D_j(x) \notin [0, x]) = 0$ for any $x \geq 0$ and $j \geq 1$,*

2. *$j \in \mathbb{N}_{\geq 1} \mapsto \Pr(D_j(x) \geq a)$ is non-increasing for any $x, a \geq 0$,*

3. *$x \geq 0 \mapsto \Pr(D_j(x) \geq a)$ is non-decreasing for any $j \in \mathbb{N}_{\geq 1}$ and $a \geq 0$.*

The second condition asserts that the random variable $D_{j-1}(x)$ has first-order stochastic dominance over $D_j(x)$. Along with the first condition, reflect that the distributions of the rejected values become progressively smaller. The last condition indicates that for any integer $j \geq 1$ and non-negative real numbers $x < y$, $D_j(y)$ has a first-order stochastic dominance over $D_j(x)$, which means that, as the value of $x$ increases, so does the potential recovered value after $j$ steps.

**The decision-maker**   In the $\mathcal{D}$-prophet inequality with deterministic decay functions, we assumed that the decision-maker has full knowledge of the functions $D_1, D_2 \dots$. In the randomized setting, we assume instead that the decision-maker knows the distributions of the decay functions, i.e. knows the distribution of the random variables $D_j(x)$ for all $x \geq 0$ and $j \geq 1$. However, they do not observe their values until they decide to stop. The online selection process is therefore as follows: the algorithm knows beforehand the distributions of the decay functions, then at each step, it observes a new item with value $X_i$, and decides to stop or continue. Once they decide to stop at some time $\tau$, they observe the values $D_1(X_{\tau-1}), \dots, D_\tau(X_1)$ and then they choose the maximal one. As a consequence, the stopping time $\tau$ is independent of the randomness induced by the decay functions. As in the deterministic case, the expected reward of any algorithm $\mathsf{ALG}$ can be written as

$$\mathbb{E}[\mathsf{ALG}^{\mathcal{D}}(X_1, \dots, X_n)] = \mathbb{E}[\max_{0 \leq i \leq \tau-1}\{D_i(X_{\tau-i})\}] .$$

**The limit decay**    A key result in our paper is the reduction of the problem to the case where all the decay functions are identical, and we prove this reduction by considering the pointwise limit of the decay functions. In the case of random decay functions, instead of the pointwise convergence, it holds for all $x \geq 0$ that the random variables $(D_j(x))_j$ converge in distribution to some random variable $D_\infty(x)$. In fact, for any $x \geq 0$ and $a \geq 0$, the sequence $(\Pr(D_j(x) \geq a))_{j \geq 1}$ is non-increasing and non-negative, thus it converges to some constant $G(x, a)$. Given that $x \mapsto \Pr(D_j(x) \geq a)$ is non-decreasing for any $j$, we obtain by taking the limit $j \to \infty$ that $x \mapsto G(x, a)$ is non-increasing, and with similar argument we obtain, for any $x \geq 0$, that $G(x, a) = 1$ for all $a \leq 0$ and $G(x, a) = 0$ for all $a > x$. Therefore, $a \mapsto 1 - G(x, a)$ defined the cumulative distribution of a random variable $D_\infty$ such that

- $x \geq 0 \mapsto \Pr(D_\infty(x) \geq a)$ is non-decreasing for all $a \geq 0$,
- $\Pr(D_\infty(x) \notin [0, x]) = 0$ for al $x \geq 0$.

Therefore, for all $x \geq 0$, $D_\infty(x)$ is the limit in distribution of $(D_j(x))_j$, hence a sequence $\mathcal{D}' = (D_1', D_2', \ldots)$ of mutually independent random processes such that $D_j'(x) \sim D_\infty(x)$ for any $j \geq 1$ and $x \geq 0$ defines a sequence of decay functions. We say in this case that all the decay functions are identically distributed as $D_\infty$. Moreover, it holds for all $x \geq 0$ that $\mathbb{E}[D_\infty(x)] = \lim_{j \to \infty} \mathbb{E}[D_j(x)] = \inf_{j \geq 1} \mathbb{E}[D_j(x)]$

From there, all the proofs of Section 2 can be easily generalized to the case of random decay functions, and it follows that we can restrict ourselves to studying identically distributed decay functions. Moreover, Proposition 3.2 can be generalized to the case of random decay functions, and the necessary condition for surpassing $1/2$ becomes $\inf_{x>0} \frac{\mathbb{E}[D_\infty(x)]}{x} > 0$. Similarly, using that the stopping time $\tau$ of the algorithm is independent of randomness induced by $D_\infty$, Proposition 3.3 remains true with $\gamma = \inf_{x>0} \frac{\mathbb{E}[D_\infty(x)]}{x}$.

**Lower bounds**    For establishing lower bounds, observe that, for any random decay functions $\mathcal{D}$, if we denote $H_j(x) = \mathbb{E}[D_j(x)]$ for all $x$, then $\mathcal{H} = (H_1, H_2, \ldots)$ defines a sequence of deterministic decay functions. Furthermore, for any instance $X_1, \ldots, X_n$ and any algorithm ALG, it holds that

$$
\begin{aligned}
\mathbb{E}[\mathsf{ALG}^{\mathcal{D}}(X_1, \ldots, X_n)] &= \mathbb{E}[\max_{0 \leq i \leq \tau - 1} \{D_i(X_{\tau - i})\}] \\
&= \mathbb{E}\Big[\mathbb{E}[\max_{0 \leq i \leq \tau - 1} \{D_i(X_{\tau - i})\} \mid \tau, X_1, \ldots, X_n]\Big] \\
&\geq \mathbb{E}\Big[\max_{0 \leq i \leq \tau - 1} \big\{\mathbb{E}[D_i(X_{\tau - i}) \mid \tau, X_1, \ldots, X_n]\big\}\Big] \\
&= \mathbb{E}\Big[\max_{0 \leq i \leq \tau - 1} \{H_i(X_{\tau - i})\}\Big] \\
&= \mathbb{E}[\mathsf{ALG}^{\mathcal{H}}(X_1, \ldots, X_n)].
\end{aligned}
$$

It follows that lower bounds established for deterministic decay functions can be extended to random decay functions by considering their expectations.

**Implications**    With the previous observations, both the lower and upper bounds, depending on $\gamma_{\mathcal{D}}$ that we proved in the deterministic $\mathcal{D}$-prophet inequality can be generalized to the random $\mathcal{D}$-prophet inequality, by taking

$$
\gamma_{\mathcal{D}} = \inf_{x>0} \inf_{j \geq 1} \frac{\mathbb{E}[D_j(x)]}{x} \; .
$$

