# OpenReview forum: "Lookback Prophet Inequalities"
_NeurIPS.cc/2024/Conference — NeurIPS 2024 poster_

### Official Review · Reviewer_g6ab · 2024-07-11

**Soundness:** 3
**Presentation:** 3
**Contribution:** 2
**Rating:** 5
**Confidence:** 3

**Summary:**

This paper extends the standard online selection problem by enabling the decision-maker to choose previous items with some discount that is captured by decay functions. The authors analyze the competitive ratio for different observation orders by giving a reduction from general decay functions to simple decay function.

**Strengths:**

1. This paper extends the standard prophet inequality to a more general and realistic form.
2. This paper gives a general reduction that characterizes the property of decay functions.
3. The competitive ratio for adversarial order is tight.

**Weaknesses:**

1. The analyses for random order and IID cases are not ideal. It seems neither upper nor lower bounds are tight for general $\gamma$.
2. I am not sure about the technical contribution of this paper. What is the biggest technical challenge in the analysis?

**Questions:**

1. What is the biggest technical challenge in the analysis?
2. I feel it is better to emphasize the choice of distribution with support ${0,a,b}$ when proving the upper bound of $\gamma$-prophet inequality (maybe change the statement of theorems). This is because $CR^D(ALG) \le \sup_A \frac{\mathbb{E}[A^{\gamma_D}(I)]}{\mathbb{E}[OPT(I)]}$ for $I$ taking values in $0,a,b$ and $\sup_A CR^{\gamma_D}(A) =  \sup_A \inf_I \frac{\mathbb{E}[A^{\gamma_D}(I)]}{\mathbb{E}[OPT(I)]} \le f(\gamma_D)$ (current statement in Theorem 4.4/4.5) does not directly imply $CR^D(ALG) \le f(\gamma_D)$.

**Limitations:**

Yes.

---

> ### Author Rebuttal · Authors · 2024-08-03
>
> We thank the reviewer for the time and effort spent on our submission. Below we address the raised concerns and questions.
> ### Weaknesses
> * **Upper bounds.** The reduction from the $D_\infty$- to the $\gamma$-prophet inequality relies on using distributions with support {$0,a,b$}, which is a limitation. Proving better upper bounds would require improving the reduction itself. We are not sure if this is feasible with arbitrary decay functions
> * **Lower bounds.** The algorithms we provide for the IID and random order models match the optimal single-threshold algorithms for $\gamma = 0$. Better algorithms require using time-dependent thresholds, which are technically difficult to analyze even for $\gamma = 0$. Multiple prior works on prophet inequalities have studied either the IID or the random order model (see the related work section), incrementally improving the lower and upper bounds. Moreover, finding an optimal algorithm in the prophet inequality with random order is still an open question.
>
> ### Questions
> * **Technical challenges.** The first big technical challenge lies in the reduction from the $\mathcal{D}$- to the $D_\infty$-prophet inequality, as we consider very general classes of decay functions with minimal assumptions. Moreover, the reduction relies on explicitly constructing hard instances for which the decay functions $\mathcal{D}$ and $D_\infty$ are equivalent, which requires very adequate concentration inequalities and rigorous tuning of the parameters in the construction. Moreover, a separate proof is given for each order model, as different arguments are required. The second big technical challenge lies in analyzing the algorithms for the $\gamma$-prophet inequality. With single-threshold algorithms, as shown in Equation (4), there is an additional term of $E[\gamma (\max X_i) 1_{\max X_i<\theta}]$ compared to the case of $\gamma=0$, which cannot be lower bounded by $c E[\max X_i]$ for a universal constant $c$ independent of the distributions. Hence manipulating this term to improve upon the competitive ratio of the case $\gamma = 0$ is challenging. Proving upper bounds is also more technical than the case $\gamma = 0$.
> * **Statement of upper bounds in the $\gamma$-prophet inequality.** We agree with the reviewer and thank them for the suggestion. There is a discussion on this point in Section 4.4. However, we will revise the statements of Theorems 4.3, 4.4, and 4.6 to place greater emphasis on this aspect.

---

> > ### Comment · Reviewer_g6ab · 2024-08-10
> >
> > I thank the authors for the response. I do not have further questions now.

---

> > > ### Author Response · Authors · 2024-08-11
> > >
> > > We thank the reviewer for getting back to us. We understand from their response that we have adequately addressed the concerns raised in their review. If this is the case, we hope the reviewer will consider raising their rating to reflect their satisfaction with our response. Otherwise, we are more than willing to engage in further discussion and provide the necessary clarifications.

---

### Official Review · Reviewer_Q5gj · 2024-07-13

**Soundness:** 4
**Presentation:** 3
**Contribution:** 2
**Rating:** 6
**Confidence:** 3

**Summary:**

**Problem Studied**

This paper introduces the problem of "lookback prophet inequalities", which is a variant of prophet inequalities. In this problem, at the stopping time, instead of always selecting the current item, the algorithm is allowed to select any item that has arrived up to now. However, items in the past have their value discounted by known decay functions. The goal is to design algorithms with a good competitive ratio, which is the ratio of the algorithm's expected value to the expectation of the highest-value item in hindsight.

**Main Results / Contributions**

The first main contribution of this paper is to define the problem of lookback prophet inequalities. The paper considers the problem in the adversarial order, random order, and IID models. It turns out that for all three models, when analyzing the worst-case competitive ratio, one can assume that the decay functions are all of the form $x \to \gamma x$, for some $\gamma$. For the adversarial model, matching upper and lower bounds are derived.  For the random order and IID models, upper and lower bounds are derived, but they do not match.

**Strengths:**

The paper is generally well-written. The definitions and theorem statements are clear, and the proof sketches in the first 9 pages of the paper gives the reader a good intuition about why the statements are true. The paper introduces a new problem (lookback prophet inequalities), which to my knowledge has not be studied before. The problem definition is clean and I think it is a reasonably natural formulation.

**Weaknesses:**

There are already numerous papers on variants of the prophet inequality, and this paper introduces yet another one. At this point, one must question how interesting or significant this contribution is. However, the problem definition appears reasonable, and the proofs seem to be fairly clean, which is why I am recommending a weak acceptance.

**Questions:**

1. I think the real estate example doesn't make too much sense. If the seller returns to a previous buyer, and it turns out the buyer is no longer interested, the seller can continue to sell the house.

2. The restaurant example also seems a bit strange to me. If the cost of revisiting a restaurant depends on the distance you need to walk back to it, this doesn't seem to be capturable with the current definitions of the decay functions, no? (Currently, the decay functions only model how many steps back into the past you look. E.g. If you are going from restaurant 3 to restaurant 1, this uses the same decay function as going from restaurant 5 to restaurant 3.)

3. It seems like the reduction to $\mathcal{D}_\infty$ relies on the fact that $n$ goes to infinity. Are there better competitive ratios that are possible for a fixed $n$?

4. There are several typos in the paper, notably in the title ("Inequalitites" -> "Inequalities").

**Limitations:**

Yes.

---

> ### Author Rebuttal · Authors · 2024-08-03
>
> We thank the reviewer for the time and effort spent on our submission. Below we address the raised concerns and questions.
>
> ### Weakness
> Indeed, many variants of the prophet inequality are used to model specific use cases. For example, 'Learning Online Algorithms with Distributional Advice' (ICML 2021) explores a situation where a decision-maker with access only to samples competes against an adversary with access to distributions, and 'Fairness and Bias in Online Selection' (ICML 2021) examines the case of items belonging to distinct, incomparable groups.
> On the other hand, the model we study is a generalization of the prophet inequality, which removes the overly pessimistic assumption of irrevocable rejection decisions, and captures much more general and more realistic online selection problems, which is why believe it is a highly relevant model.
>
> ### Questions
> * **Real estate example.** In the introduction, we use examples to illustrate that rejection decisions can often be reversed in practical scenarios. While we acknowledge the reviewer's point that the lookback prophet inequality does not perfectly model a real estate scenario, we believe it offers a more realistic representation than the standard prophet inequality, and constitutes an important step towards closing the gap between theory and practice in online selection problems.
> * **Restaurant example.** Imagine restaurants lined up along a street, with roughly equal distances between them. If there is a long distance between two consecutive restaurants, then it can be represented by empty restaurant spots, each with a zero reward. This can be modeled with decay functions of the form $D_j = cj$ with $c$ a constant. In a more general scenario, restaurants could be located anywhere on a map, which can be modeled as a decision-maker traversing a path in an arbitrary metric space. This problem is not captured by the lookback prophet inequality, and exploring it would be an interesting direction for future research.
> * **Competitive ratio for fixed $n$.** The competitive ratio is defined as the worst-case ratio over instances of all sizes, this is why considering arbitrarily large instances for proving upper bounds is not a limitation. Nonetheless, for the random order and IID models, we believe that better ratios are possible for a fixed $n$, as suggested for example in Lemma 4.5. Some prior works, such as "Comparisons of stop rule and supremum expectations of iid random variables", focused on characterizing the optimal ratios for fixed $n$ in the IID model. The authors use dynamic programming to give a recursive characterization of these ratios, but the computations are cumbersome and highly technical, and yet they fail to give closed-form expressions or precise estimations. Investigating this question in the Lookback Prophet inequality with arbitrary decay functions is surely a very interesting research direction, but also very challenging.
> * **Typos.** We thank the reviewer for their remark. We have reviewed the paper in detail and corrected all the typos.

---

> > ### Comment · Reviewer_Q5gj · 2024-08-12
> >
> > Thank you for your response, I appreciate it!

---

### Official Review · Reviewer_us2g · 2024-07-13

**Soundness:** 3
**Presentation:** 4
**Contribution:** 3
**Rating:** 6
**Confidence:** 3

**Summary:**

This paper studies a variant of prophet inequalities where the agent can reuse a previous item at a decaying price. Analyzing three models, adversarial model, random order model, and iid model, this paper gives various lower and upper bounds about the competitve ratio that an algorithm may achieve in these setups.

The paper mainly follows a reduction idea. It first show that for any decaying function family $\mathcal D$, the optimal competitive ratio is close to that for the limiting decaying function $D_{\infty}$, which reduces the $\mathcal D$-prophet problem to a $D_{\infty}$ one. This idea is further generalized so that it suffices to consider linear functions (i.e., $x\mapsto \gamma x$ for some $\gamma$) as the CR for $D_{\infty}$ is again close to that of $x\mapsto (\lim_{x\to \infty} \frac{D_{\infty}(x)}{x})x$. Equipped with these reductions, the authors
1. establish an upper bound on the CR in the adversarial model, where a matching online algorithm is also proposed;
2. develop an upper bound that is more general than previous works (considering classical prophet inequalities) in the random order model, but the lower bound algorithm is neither computationally feasible nor optimal; and
3. design an algorithm in the iid case whose asymptotic CR is the same as that in the random order model.

**Strengths:**

1. The setup is new, and it looks to of importance.
1. The paper is well-written so it is easy to get the intuitions behind the reductions.
2. The result for adversarial case is optimal.

**Weaknesses:**

1. For the first reduction ($\mathcal D$ to $D_{\infty}$), it seems that only some order models mentioned in the main text allows such a reduction. Why you only consider these models, and are there any other models excluded from this reduction (or do you have any intuition on what kinds of order models can ensure the reduction)?
2. For random order models and iid models, the algorithms designed are sub-optimal and cannot be implemented easily.

**Questions:**

1. See Weakness 1
2. 'However, considering larger classes of algorithms, the competitive ratios achieved in the IID model are better than those of the random order model.' Can you give some examples? I see in the limitations you mentioned that such algorithms are not designed in your $\mathcal D$-prophet model. Why they do not generalize?

**Limitations:**

Clearly stated

---

> ### Author Rebuttal · Authors · 2024-08-03
>
> We thank the reviewer for the time and effort spent on our submission. Below we address the raised concerns and questions.
> ### Weaknesses
> * **Other models.** We studied the models where the only decision made by the algorithm is the stopping time. The only model not included in our analysis is the "order selection model", where the decision-maker additionally decides in which order to observe the samples. To prove the reduction in that model, our intuition is that hard instances can be constructed using a trick similar to the one we used for the IID model, but then the optimal order in which the samples of these instances are observed should also be characterized.
> * **Optimality.** The algorithms we provide for the IID and random order models match the optimal single-threshold algorithms when $\gamma = 0$. Better algorithms require using time-dependent thresholds, which are difficult to analyze even for $\gamma = 0$. Multiple prior works on prophet inequalities have studied either the IID or the random order model (see the related work section), incrementally improving the lower and upper bounds. Moreover, finding an optimal algorithm in the prophet inequality with random order is still an open question.
> * **Implementation.** The algorithms we propose only require computing the solution $p$ of the equation $1-(1-\gamma)p = \frac{1-p}{-\log p}$, which can be easily computed numerically with very high precision as explained in Line 307. The proposed threshold then depends on the distribution of the maximum reward, which is standard for prophet inequalities. Additionally, we give in Corollary 4.4.1 a simpler threshold that results in a slightly weaker competitive ratio.
>
> ### Questions
> * **Separation between the IID and random order models.** For $\gamma = 0$, the optimal single-threshold algorithm for both the iid and random order models has a competitive ratio of $1-1/e$. However, as mentioned in the related work section, using time-dependent thresholds enables a competitive ratio of $\approx 0.745$ in the iid model, which is optimal. On the other hand, in the random order model, no algorithm has a competitive ratio better than  $\sqrt{3}-1 \approx 0.732$, which shows the separation between the best competitive ratios in both models.
> * **Generalizing the algorithms.** A key property satisfied by single-threshold algorithms is Equation $(4)$ in line 270, on which our analysis relies. For algorithms using time-dependent thresholds, this property is no longer true (see Line 272), and different analysis techniques must be used to generalize them. Studying such algorithms is difficult and very technical even for $\gamma = 0$, and multiple separate works have focused either on the IID or the random order model to advance the analysis of such algorithms (see the related work section).

---

> > ### Comment · Reviewer_us2g · 2024-08-12
> >
> > Thanks for your response! I'd like to keep my score unchanged.

---

### Decision · Program_Chairs · 2024-09-25

**Decision:**

Accept (poster)

**Comment:**

The submission introduces a new and well-motivated variant of the Prophet Inequality.
The reviewers are generally satisfied with both the results and techniques.
I recommend accept.